# Genome-wide associations of aortic distensibility suggest causality for aortic aneurysms and brain white matter hyperintensities

Catherine M. Francis [1,2,33], Matthias E. Futschik[3,4,33], Jian Huang[3,33], Wenjia Bai [5,6], Muralidharan Sargurupremraj[7,8], Alexander Teumer [9,10,11], Monique M. B. Breteler [12,13], Enrico Petretto [14,15,16], Amanda S. R. Ho [16], Philippe Amouyel [17,18,19,20], Stefan T. Engelter[21,22], Robin Bülow[23], Uwe Völker [10,24], Henry Völzke[9,10], Marcus Dörr [10,25], Mohammed-Aslam Imtiaz[12], N. Ahmad Aziz [12,26], Valerie Lohner [12], James S. Ware [1,2,4], Stephanie Debette [8,27], Paul Elliott [3,28,29,30,31,32], Abbas Dehghan [3,28,34] ✉ & Paul M. Matthews [5,28,31,34] ✉

Aortic dimensions and distensibility are key risk factors for aortic aneurysms and dissections, as well as for other cardiovascular and cerebrovascular diseases. We present genome-wide associations of ascending and descending aortic distensibility and area derived from cardiac magnetic resonance imaging (MRI) data of up to 32,590 Caucasian individuals in UK Biobank. We identify 102 loci (including 27 novel associations) tagging genes related to cardiovascular development, extracellular matrix production, smooth muscle cell contraction and heritable aortic diseases. Functional analyses highlight four signalling pathways associated with aortic distensibility (TGF-β, IGF, VEGF and PDGF). We identify distinct sex-specific associations with aortic traits. We develop co-expression networks associated with aortic traits and apply phenome-wide Mendelian randomization (MR-PheWAS), generating evidence for a causal role for aortic distensibility in development of aortic aneurysms. Multivariable MR suggests a causal relationship between aortic distensibility and cerebral white matter hyperintensities, mechanistically linking aortic traits and brain small vessel disease.

The aorta acts as both conduit and buffer[1], conveying oxygenated blood from the heart to the systemic circulation, and dampening the pulse pressure to which peripheral circulations are subjected. Diseases affecting the aorta are common and their complications are associated with high mortality even in young people. Quantitative aortic traits (aortic dimensions and functional measures) can predict progression of these aortopathies. For example, the elastic function of the thoracic aorta and aortic dimensions are key determinants of rates of growth of thoracic aortic aneurysms[2–4]. At a population level, aortic traits are also clinically important predictors of risks of cardiovascular and cerebrovascular diseases[5–10].

The distensibility of the proximal thoracic aorta allows it to buffer pressure changes associated with cardiac ejection, acting to protect the cerebral vasculature from high pulse pressures. However, as the

aorta remodels and stiffens with age, the cerebral circulation is exposed to higher pulse pressures. This decline in elastic function may be measured as a decrease in distensibility, a factor independently predictive of cerebral microvascular disease, the development of age-related dementia and neurodegenerative changes of Alzheimer's Disease (AD)[11–13]. Recent data also have provided evidence for an association of aortic distensibility with cognitive performance in the general population[14]. White matter hyperintensities (WMH) represent the most common brain imaging feature of small vessel disease and predict mortality and morbidity with aging (including risks of stroke (ischaemic and haemorrhagic), dementia, and functional impairment[15–18]). Aortic stiffness is a stronger predictor of WMH volume than blood pressure or hypertension alone[19,20] and has effects additive to those of hypertension in predicting WMH[19–21]. The genomic bases of these relationships have not been well explored to date.

With age, aortic stiffening arises from changes in composition of the aortic wall, including degradation of the elastic fibres and decreased cellularity, along with a relative increase in the collagen content of the aorta (although the absolute amount decreases)[22,23]. In addition, extracellular matrix proteins themselves undergo conformational and biochemical changes which alter their passive mechanical properties. These remodelling processes are driven by TGF-β signalling pathways and accelerated by oxidative stress and inflammation[24] acting on the cells in the aortic wall. These cellular and molecular drivers of worsening aortic elastic function are reflected in macroscopic changes with age.

Here, we used convolutional neural networks for automated aortic segmentation[25] to measure ascending and descending aortic areas and distensibilities on cardiac magnetic resonance (MRI) images from UK Biobank, which is currently the largest cardiac imaging epidemiological study[26]. We have described our approach to derivation of imaging-derived quantitative aortic traits and the distribution of these traits in a smaller group from the same population in an earlier report[25,27]. We derived six aortic traits (ascending aortic distensibility (AAdis), descending aortic distensibility (DAdis), maximum ascending aortic area (AAmax), minimum ascending aortic area (AAmin), maximum descending aortic area (DAmax) and minimum descending aortic area (DAmin)) in up to 32,590 (depending on the specific trait) UK Biobank participants, who were free from known aortic disease We then performed a genome-wide association study (GWAS) of the six cardiac magnetic resonance (CMR)-derived aortic traits and carried out functional analyses and a series of Mendelian randomisation (MR) studies to investigate possible causal associations of the aortic measures with aortic aneurysms and brain small vessel disease. We also explored the bidirectional relationship of aortic traits with indices of blood pressure.

## Results

Cohort demographics are presented in Supplementary Data 1a and exclusions presented in Supplementary Fig. 1. The distributions of the aortic traits for the study cohort are shown in Supplementary Fig. 2a, b, and for excluded non-Caucasian participants in Supplementary Fig. 2c and Supplementary Data 1b.

### Correlations with biometric variables

Aortic traits correlated as expected with biometric variables (see Supplementary Data 1b for details). The strongest correlations with distensibilities were with age (AAdis $r = -0.552$, $p < 0.001$; DAdis $r = -0.539$, $p < 0.001$). Aortic diameters correlated most strongly with body size variables. Interestingly, the correlations with height and weight were much stronger than with BMI, e.g., for AAmax-height $r = 0.383$, AAmax-weight $r = 0.391$ and AAmax-BMI $r = 0.210$ ($p < 0.001$ for all).

### SNP-based heritability

We estimated the proportion of the variability in aortic traits that could be attributed to common genetic variation from an analysis of SNP-based heritability ($h^2_{SNP}$) using linkage disequilibrium score regression (LDSC) (Supplementary Data 2). $h^2_{SNP}$ estimates ranged from 0.10 (for DAdis single trait) to 0.41 (for AAmax). We also tested for heritability of distensibility traits using multi-trait analysis (MTAG, $h^2_{SNP} = 0.21$ for DAdis and $h^2_{SNP} = 0.24$ for AAdis).

### Phenotypic and genotypic correlations between traits

We found strong phenotypic and genotypic correlations between maximum and minimum aortic areas (phenotypic $r = 0.99$, $p < 2.2 \times 10^{-16}$; genotypic $r_g = 0.99$, $p < 1 \times 10^{-50}$ for the ascending aorta; phenotypic $r = 0.98$, $p < 2.2 \times 10^{-16}$; genotypic $r_g = 0.99$, $p < 1 \times 10^{-50}$ for the descending aorta).There were lower correlations between ascending and descending aortic traits (phenotypic $r = 0.60$, $p < 2.2 \times 10^{-16}$ and genotypic $r_g = 0.45$, $p < 1.7 \times 10^{-25}$ for the minimum aortic areas and phenotypic $r = 0.74$, $p < 2.2 \times 10^{-16}$ and genotypic $r_g = 0.45$, $p < 9.25 \times 10^{-7}$ for distensibilities) consistent with known biological and functional differences along the course of the aorta[1]. Correlations are presented in full in Supplementary Figs. 3 and 4 and Supplementary Data 3 and 4.

### Single and multi-trait aortic GWAS

Our stage 1 GWAS ($N = 32,590$ for areas and $N = 29,895$ for distensibility) identified a total of 95 significant loci (using a genome-wide significance threshold of $p < 5 \times 10^{-8}$) across the six traits, 94 of which are autosomal with one localised to the X chromosome. Genomic inflation was within acceptable limits for all traits ($\lambda = 1.147$ for the area traits; $\lambda = 1.047$ for the distensibility traits). We took advantage of the correlation between the aortic traits to enhance the power for the discovery of loci by performing the multi-trait analysis (MTAG)[28] as the second stage of our GWAS. Use of MTAG combining all six phenotypes increased the number of significant loci for the distensibility traits from 10 to 26 for the ascending aorta and from 7 to 13 for the descending aorta (Table 1), and the total number of significant loci across all aortic traits to 102. Figure 1a, b shows the Manhattan and QQ plots from GWAS of ascending and descending aortic minimum areas, which overlap almost completely with the findings for the corresponding maximum areas. Figure 1c, d shows the results of the MTAG analysis of ascending and descending aortic distensibilities. GWAS summary statistics from 9,753,033 variants with a minor allele frequency (MAF) ≥ 0.01 for the stage 1 and stage 2 (MTAG) analyses are shown in Supplementary Figs. 5–8. Significant associations for a single trait and stage 2 MTAG analyses are shown in Supplementary Data 5a–f, 6a–f, 7 and 8.

Individual loci were annotated with *cis*-expression quantitative trait loci (eQTL) and splice quantitative trait loci (sQTL) data from GTEx v8[29]. Twenty-four of the 38 loci associated with distensibilities had lead SNPs which were significant eQTLs or sQTLs for nearby genes in arterial tissue (see Supplementary Data 9 and 10 for details of eQTL, sQTL and further annotations).

The most significant associations with ascending aortic distensibility were: rs7795735, 12.6 kilobases upstream of *ELN*, a gene encoding elastin; rs201281936, which is in a lncRNA (*CTD-2337A12.1*) 119 kilobases 3' of *PCSK1* (Proprotein Convertase Subtilisin/Kexin Type 1); rs57130712, which is in a locus spanning *SMG6* (SMG6 Nonsense Mediated MRNA Decay Factor) and *SRR* (Serine Racemase), and which is the most significant eQTL for SRR in arterial tissue ($p = 2 \times 10^{-20}$, normalised effect size (NES) $= -0.39$)[29]; and rs34557926, an intronic variant in *HAS2* (Hyaluronan Synthase 2).

The strongest associations with descending aortic distensibility were different. The most significant association was with rs61886305, an intronic variant in *PLCE1 (*Phospholipase C Epsilon 1). This lead SNP is a strong eQTL for *PLCE1* in arterial tissue ($p = 1.1 \times 10^{-8}$,

**Table 1 | Significant associations with distensibility in single and multi-trait (MTAG) genome-wide analyses**

| Locus | Lead SNP rsID | Chr | BP | Annotation | Closest gene | Effect allele | Non-effect allele | EAF | No. of lead SNPs | Beta | P | Trait association reaching genome-wide significance |
|---|---|---|---|---|---|---|---|---|---|---|---|---|
| 1* | rs835341 | 1 | 53064012 | intergenic | GPX7 | A | G | 0.47 | 1 | 0.027 | 7.78E-09 | AAdis |
| 2 | rs824510 | 2 | 19725556 | intergenic | AC010096.2 | G | A | 0.68 | 1 | 0.037 | 3.73E-14 | AAdis |
| 3 | rs9306895 | 2 | 20878153 | ncRNA_exonic | AC012065.7: RP11-130L8.1 | T | C | 0.64 | 1 | 0.038 | 4.87E-17 | DAdis |
| 4$ | rs6724315 | 2 | 46363699 | intronic | PRKCE | T | C | 0.87 | 1 | -0.053 | 3.00E-08$ | DAdis |
| 5 | rs35303331 | 2 | 164921770 | ncRNA_intronic | AC092684.1 | A | G | 0.77 | 1 | 0.031 | 2.04E-08 | AAdis |
| 6# | rs541051407 | 3 | 41871295 | intronic | ULK4 | G | C | 0.88 | 1 | -0.058 | 2.20E-08# | AAdis |
| 7* | rs7638565 | 3 | 64723072 | ncRNA_intronic | ADAMTS9-AS2 | A | G | 0.40 | 1 | 0.026 | 4.53E-08 | AAdis |
| 8 | rs55914222 | 3 | 128202943 | intronic | GATA2 | G | C | 0.97 | 1 | -0.086 | 8.48E-10 | AAdis |
| 9 | rs79957887 | 3 | 187006335 | intronic | MASP1 | C | T | 0.92 | 1 | 0.054 | 3.14E-11 | DAdis |
| 10 | rs17020769 | 4 | 146800922 | intronic | ZNF827 | C | T | 0.52 | 1 | 0.025 | 4.00E-08 | AAdis |
| 11* | rs13158444 | 5 | 51201361 | intergenic | RNA5SP182 | T | C | 0.40 | 1 | -0.027 | 9.61E-09 | AAdis |
| 12 | rs79051849 | 5 | 81848970 | intergenic | CTD-2015A6.1 | A | G | 0.81 | 1 | -0.039 | 5.29E-11 | AAdis |
| 13 | rs201281936 | 5 | 95606717 | ncRNA_intronic | CTD-2337A12.1: RP11-254I22.2 | A | C | 0.64 | 2 | -0.053 | 4.79E-28 | AAdis |
| 14 | rs337101 | 5 | 122550646 | intergenic | PRDM6 | T | C | 0.72 | 2 | 0.034 | 2.93E-11 | AAdis |
| 15 | rs2490445 | 6 | 12544672 | intergenic | RPL15P3 | T | C | 0.70 | 1 | 0.028 | 2.29E-08 | AAdis |
| 16* | rs1406667 | 6 | 122178493 | downstream | HMGB3P18 | A | G | 0.90 | 1 | 0.043 | 3.53E-09 | DAdis |
| 17 | rs56072713 | 7 | 73427710 | intergenic | ELN | G | A | 0.54 | 1 | 0.034 | 1.83E-15 | DAdis |
| 17 | rs7795735 | 7 | 73429482 | intergenic | ELN | T | A | 0.55 | 4 | 0.077 | 4.78E-62 | AAdis |
| 18 | rs9721183 | 8 | 75781818 | intergenic | RP11-758M4.4 | C | T | 0.63 | 1 | -0.033 | 1.39E-11 | AAdis |
| 19* | rs7832313 | 8 | 92198211 | intronic | LRRC69 | G | A | 0.31 | 1 | 0.026 | 1.48E-08 | DAdis |
| 20 | rs11992999 | 8 | 122639443 | intronic | HAS2 | T | C | 0.71 | 1 | -0.038 | 4.42E-14 | AAdis |
| 21 | rs34557926 | 8 | 124607159 | intergenic | RN7SKP155 | C | T | 0.64 | 1 | 0.039 | 7.75E-16 | AAdis |
| 21 | rs7006122 | 8 | 124608614 | intergenic | RN7SKP155 | T | G | 0.64 | 2 | 0.030 | 1.54E-11 | DAdis |
| 22 | rs9702161 | 10 | 30077914 | intergenic | SVIL | T | C | 0.29 | 1 | -0.029 | 1.52E-08 | AAdis |
| 22 | rs7096778 | 10 | 30165983 | intergenic | RP11-224P11.1; SVIL | T | C | 0.41 | 2 | -0.030 | 3.75E-12 | DAdis |
| 23$ | rs10857614 | 10 | 49829983 | intronic | ARHGAP22 | T | C | 0.52 | 1 | 0.034 | 4.10E-08$ | AAdis |
| 24 | rs10761716 | 10 | 64882300 | intergenic | RP11-144G16.1 | C | G | 0.56 | 1 | -0.026 | 4.34E-08 | AAdis |
| 25 | rs78629306 | 10 | 95897188 | intronic | PLCE1 | G | C | 0.82 | 2 | 0.039 | 2.76E-10 | AAdis |
| 25 | rs61886305 | 10 | 95902053 | intronic | PLCE1 | C | A | 0.83 | 6 | 0.051 | 2.49E-19 | DAdis |
| 26 | rs875106 | 11 | 70005641 | intronic | ANO1 | G | A | 0.48 | 2 | -0.027 | 8.77E-09 | AAdis |
| 27 | rs11046213 | 12 | 22008367 | ncRNA_intronic | ABCC9:RP11-729I10.2 | G | T | 0.59 | 1 | -0.032 | 1.18E-11 | AAdis |
| 28* | rs61927702 | 12 | 33631695 | intergenic | RP11-56H16.1 | A | G | 0.47 | 1 | -0.026 | 2.87E-09 | DAdis |
| 29 | rs12863716 | 13 | 22862729 | intergenic | MTND3P1 | C | T | 0.78 | 1 | -0.042 | 1.23E-13 | AAdis |
| 30* | rs8014161 | 14 | 92393198 | intronic | FBLN5 | T | A | 0.64 | 1 | -0.032 | 1.91E-12 | DAdis |
| 31 | rs1441358 | 15 | 71612514 | intronic | THSD4 | T | G | 0.66 | 1 | -0.032 | 6.42E-11 | AAdis |
| 32 | rs77870048 | 16 | 69965021 | intronic | WWP2 | C | T | 0.95 | 1 | -0.066 | 1.59E-10 | AAdis |
| 33* | rs3851734 | 16 | 75371920 | intronic | CFDP1 | T | C | 0.41 | 1 | -0.029 | 2.98E-13 | DAdis |
| 34# | rs2228685 | 16 | 83065965 | intronic | CDH13 | T | A | 0.46 | 1 | 0.037 | 2.80E-09# | AAdis |
| 35 | rs28375406 | 16 | 88996841 | intronic | CBFA2T3 | A | G | 0.63 | 1 | -0.026 | 3.88E-08 | AAdis |

**Table 1 (continued) | Significant associations with distensibility in single and multi-trait (MTAG) genome-wide analyses**

| Locus | Lead SNP rsID | Chr | BP | Annotation | Closest gene | Effect allele | Non-effect allele | EAF | No. of lead SNPs | Beta | P | Trait association reaching genome-wide significance |
|---|---|---|---|---|---|---|---|---|---|---|---|---|
| 36 | rs57130712 | 17 | 2089035 | intronic | SMG6 | A | G | 0.69 | 3 | 0.042 | 3.46E-17 | AAdis |
| 36 | rs1532292 | 17 | 2097483 | intronic | SMG6 | T | G | 0.62 | 1 | 0.027 | 5.57E-10 | DAdis |
| 37* | rs112009052 | 19 | 41099501 | intronic | LTBP4 | T | A | 0.99 | 1 | −0.105 | 2.46E-08 | DAdis |
| 38* | rs28451064 | 21 | 35593827 | ncRNA_intronic | AP000320.7: AP000318.2 | G | A | 0.87 | 1 | −0.039 | 1.39E-08 | AAdis |

Summary statistics are shown for lead SNPs which were genome-wide significant ($p < 5 \times 10^{-8}$) in MTAG analysis apart from the four loci identified by # or $ which were significant in single-trait analysis only. LD information for the lead SNPs at these four loci is provided in Supplementary Data 8. # = genome wide significant in only the single-trait analysis, not the multi-trait analysis, but also significantly associated with aortic area traits. $ = genome wide significant in only the single-trait analysis of distensibility and in no other aortic traits. *=novel locus not identified in previous GWAS of ascending or descending aortic traits. Where lead SNPs are listed separately under the same locus number, different lead SNPs are listed separately under the same locus number. *MAF* minor allele frequency, *Chr* chromosome, *BP* position (GRCh37); *P* unadjusted *p* value of association (mixed model association implemented in BOLT-LMM[69] for shaded loci; multi-trait association analysis implemented in MTAG[28] for all other loci).

NES = −0.20)[29]. The next strongest association was with rs9306895, an intronic variant in *GDF7* (Growth Differentiation Factor 7), which is a strong eQTL for both *GDF7* and *LDAH* (Lipid Droplet Associated Hydrolase) in aorta[29] (*GDF7*: $p = 3.6 \times 10^{-9}$, NES = 0.20; *LDAH*: $p = 7.1 \times 10^{-28}$, NES = 0.53). A locus spanning *ELN* was associated with DAdis, with the lead SNP <1.8 kb away from the lead SNP for AAdis at this locus, and in strong LD with it ($R^2 = 0.96$; $D' = 1$). A locus in *FBLN5* (lead SNP rs8014161) was associated with descending, but not ascending distensibility.

There were four loci associated with genome-wide significance with aortic distensibilities which lost genome-wide significance in the MTAG analysis (three associated with AAdis and one with DAdis; see Table 1 for details). Two of these were not significantly associated with any other aortic traits: rs6724315 in *PRKCE* and rs10857614 in *ARHGAP22*. The latter is a strong eQTL for ARHGAP22 in aorta ($p = 2.6 \times 10^{-46}$, NES = 0.58), providing additional evidence for its biological relevance.

We compared our association results for aortic areas with those reported for aortic diameters in recent papers and preprints[30–32] based on the same UK Biobank imaging data set, but using different methods and metrics for aortic dimensions (see Supplementary Data 11). We replicated 75 of the previously reported genome-wide significant association loci and added associations for clinically relevant phenotypes (ascending and descending aortic distensibility) to identify a further 27 novel associations. Inspection of the loci for SNPs that were significant in the analysis by Pirruccello et al[30] and Benjamins et al.[32] but not in our own, generally showed SNP *p* value signals near the genome-wide significance threshold ($p < 5 \times 10^{-8}$) in our analysis. The small differences between associations in the studies could arise from differences in the methods used to generate quantitative phenotypes or from the differences in participant exclusions between the two studies (for example, we excluded data from participants with known diagnoses of aortic disease and or those who were extreme phenotypic outliers).

Novel aortic loci associated with AAdis included lead SNPs rs7638565 near *ADAMTS9*, which is a significant eQTL for this gene in aorta ($p = 5.7 \times 10^{-7}$, NES = −0.22) and rs835341, intronic in *GPX7* and a strong eQTL in the aorta ($p = 3 \times 10^{-95}$, NES = −0.92). A novel locus associated with descending aortic distensibility included lead SNP rs112009052, an sQTL for *LTBP4* in fibroblasts ($p = 9.6 \times 10^{-67}$, NES = 2.9) but not in aortic tissues that has relevance here due to this gene's role in the TGF-β pathway.

We identified 21 novel associations with aortic areas including lead SNPs in *KALRN* and *COL21A1* associated with ascending aortic areas and SNPs at loci tagging *AFAP1, FGF5/BMP3, NOX4, FES* and *GATA5/LAMA5* associated with descending aortic areas (see Supplementary Fig. 5 and Supplementary Data 5(a–f)).

### Replication
We looked up our lead SNPs for associations with aortic areas in an independent replication data set from the Study of Health in Pomerania (SHIP; N = 2787)[33]. More than 89% of the lead SNPs available for lookups in the replication data showed directionally consistent associations with aortic areas. More than 23% of these lead SNPs reached at least nominal significance (see Methods and Supplementary Data 7a–d for full replication results).

### Sex-specific aortic trait GWAS analyses
We undertook sex-specific GWAS analysis of the area phenotypes (see Supplementary Data 12). We did not perform these analyses for distensibility phenotypes due to a lack of power with the smaller cohort sizes. We contrasted associations discovered for the men and women (numbers of whom were well-balanced in the cohort) using a *z* test. There were 18 loci (ten for AAmin and eight for DAmin) at which the differences between sexes were significant (adjusted

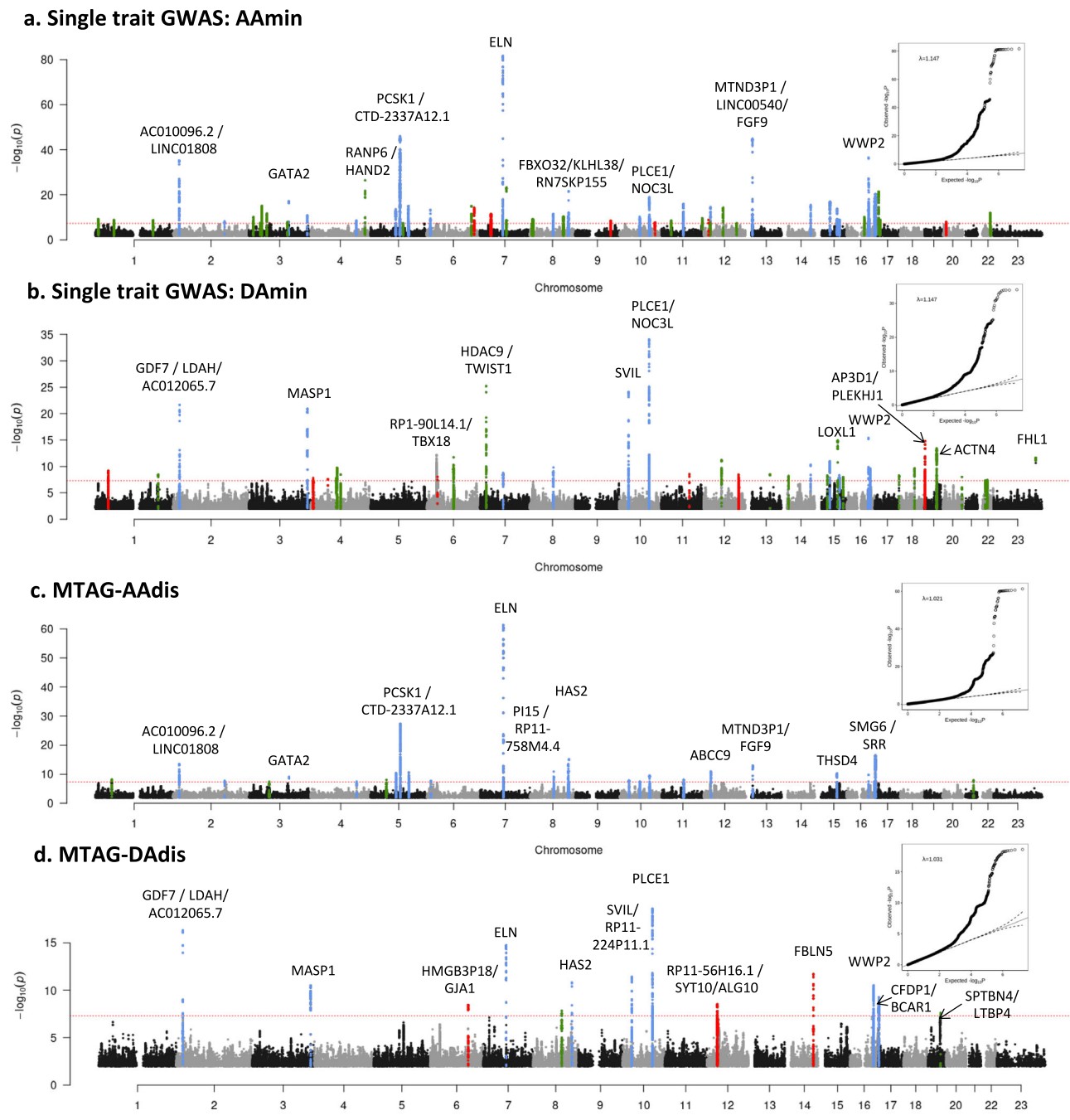

**Fig. 1 | Manhattan plots of summary statistics from GWAS of aortic traits.**
**a** Single-trait analysis of ascending aortic minimum area (AAmin) and **b** single-trait analysis of descending aortic minimum area (DAmin). Genomic inflation ($\lambda$) = 1.147 (Aamin and Damin). The y axis shows the negative log of the unadjusted p value of association (mixed model association implemented in BOLT-LMM[89]). **c** Multi-trait analysis (MTAG) of ascending aortic distensibility (AAdis) and **d** Multi-trait analysis of descending aortic distensibility (DAdis). The y axis shows the unadjusted p value of association (using MTAG[28] as discussed in Methods); negative log scale. Twenty-six association signals were identified in multi-trait analysis of ascending aortic distensibility (AAdis) and thirteen in multi-trait analysis of descending aortic distensibility (DAdis). All six traits (maximum and minimum areas and distensibility in ascending and descending aorta) were used for the MTAG analysis. Genomic inflation ($\lambda$) = 1.021 (AAdis) and 1.031 (DAdis). All panels: Red dashed lines show the genome-wide significance threshold of $P = 5 \times 10^{-8}$. Annotations of selected loci show the nearest gene and additional manual annotation of likely candidate gene(s) at the locus where appropriate. Blue: locus is genome-wide significant in multiple aortic traits, green: locus is genome-wide significant only in the corresponding trait and with nominal significance ($p < 0.01$) in other traits, red: genome-wide significant only in the corresponding trait without even nominal significance in other traits. QQ plots are shown as inserts in corresponding panels.

$p < 0.05$; see Supplementary Data 12). Seven of these associated loci were not significant even at $p < 0.05$ for one sex, despite reaching genome-wide significance ($p < 5 \times 10^{-8}$) for the other (see Table 2). Amongst these sex-specific loci were rs28699256, a missense variant in *ADAMTS7* associated with ascending aortic area in females, but not males (see Table 2; z test for sex difference, $p = 7.6 \times 10^{-4}$). This

variant in *ADAMTS7* is in strong LD with the lead SNP associated with AAmin in the full cohort at genome-wide significance (rs7182642; $R^2 = 0.76$, $D' = 0.94$). Amongst the other sex-specific signals were four others with functional data supporting potential biological roles in the aorta, all of which were significant only in males: rs72765298 in *SCAI*, a strong eQTL in aorta (see Table 2; p value for eQTL in

**Table 2 | Lead SNPs at sex-specific loci**

| SNP | Gene | Trait | Effect allele | Males | | | Females | | | Comparison | | |
|---|---|---|---|---|---|---|---|---|---|---|---|---|
| | | | | *pval* | beta | SE | *pval* | beta | SE | sig | z stat | z.pval |
| rs72765298 | *SCAI* | AAmin | T | 7.10E-09 | 15.54 | 2.68 | 0.43 | 1.77 | 2.25 | M | 5.58 | 2.43E-08 |
| rs28699256 | *ADAMTS7* | AAmin | T | 0.066 | −3.32 | 1.81 | 4.10E-09 | −8.92 | 1.52 | F | 3.37 | 7.63E-04 |
| rs632650 | *ACAD10* | DAmin | G | 1.30E-09 | −6.76 | 1.11 | 0.54 | −0.52 | 0.85 | M | −6.33 | 2.39E-10 |
| rs6573268 | *CCDC175* | DAmin | G | 1.10E-08 | 5.15 | 0.90 | 0.29 | 0.72 | 0.69 | M | 5.53 | 3.22E-08 |
| rs35346340 | *FES* | DAmin | G | 8.10E-09 | 4.93 | 0.86 | 0.081 | 1.14 | 0.65 | M | 5.00 | 5.62E-07 |
| rs9449999 | *TBX18* | DAmin | A | 2.70E-08 | −4.52 | 0.81 | 0.12 | −0.96 | 0.62 | M | −4.94 | 7.64E-07 |
| rs577351796 | *TBC1D12* | DAmin | C | 0.11 | −6.50 | 4.08 | 6.00E-11 | −19.90 | 3.04 | F | 3.74 | 1.83E-04 |

SNPs are shown if they reach genome-wide significance ($p < 5 \times 10^{-8}$) in one sex and are not significant ($p > 0.05$) in the other. *Gene* nearest gene. *Trait* aortic trait with which association in one sex is genome-wide significant, *pval* unadjusted $p$ value from sex-specific GWAS using BOLT-LMM as described in Methods, *beta* effect size from sex-specific GWAS, *SE* standard error from sex-specific GWAS, *sig* which sex the SNP has reached genome-wide significance in, *z stat* z statistic (two-tailed) for comparison between sexes, *z.pval* unadjusted $p$ value of sex comparison. All SNPs shown are significantly different between the sexes after multiple testing correction (see Supplementary Data 12 for more detailed results).

aorta = $8.9 \times 10^{-18}$, NES = −0.48); rs632650, a significant eQTL for *ALDH2* in aorta (see Table 2; $p$ value for eQTL in aorta = $3.3 \times 10^{-12}$, NES = 0.26); rs6573268, associated with DAdis, in *CCDC175* which is a significant eQTL for this gene and others in the aorta (see Table 2; $p$ value for eQTLs in aorta: for *CCDC175* $p = 1.2 \times 10^{-6}$, NES = 0.35; for *RTN1* $p = 2.4 \times 10^{-7}$, NES = 0.33 and for *L3HYPDH*, $p = 1.1 \times 10^{-4}$, NES = 0.25); and rs35346340 in *FES*, a strong eQTL for this gene in aorta (see Table 2; $p$ value for eQTL in aorta $2.4 \times 10^{-15}$, NES = 0.3).

An association with rs12663193 (intronic in *ESR1*) reached genome-wide significance only in females. However, the sex difference was not significant ($z$ test $p$ value = 0.06).

## Gene-based analysis and tissue specificity

We prioritised potentially causal genes at significant loci using two complementary strategies: FUMA[34], which integrates positional mapping, eQTL associations and HiC-derived 3D chromatin interactions (see Methods) and MAGMA[35], which aggregates SNP associations within genes. In total, FUMA identified 973 candidate genes across the six phenotypes, including 390 protein-coding genes, 164 pseudogenes, 129 lincRNAs, 115 antisense RNAs and 46 miRNAs (Supplementary Data 13, Supplementary Fig. 9). MAGMA identified 391 candidate genes with an FDR < 0.01 (Supplementary Data 14, Supplementary Fig. 10). The most significant gene associations (MAGMA) for ascending and descending aortic distensibilities are shown in Table 3.

Four genes (*MASP1, PI15, PLCE, TBC1D12* [the last likely tagging the *PLCE1* locus]) reached significance for all six aortic traits, with *ELN* at genome-wide significance in all traits except for DAmax, where it was just below the genome-wide significance threshold.

Tissue specificity analysis in MAGMA for genes associated with each phenotype demonstrated that these were significantly enriched for expression in the aorta and in the coronary artery ($p$ value for enrichment < $1 \times 10^{-3}$ in all traits), supporting the validity of our results. See Supplementary Fig. 11 for further details.

## Gene set enrichment and pathways analyses

The GO terms identified by MAGMA (Supplementary Fig. 12, Supplementary Data 15a) that were most significantly associated with our aortic phenotypes highlighted processes important for the development of aortic aneurysms and dissection, such as "extracellular matrix structural constituent" and "smooth muscle contraction", as well as GO terms related to cardiovascular development.

DEPICT implicated similar ontologies and identified three molecular pathways significantly enriched in our data (FDR < 0.01) for at least one aortic trait (see Fig. 2; Supplementary Data 15b) and of nominal significance in all other traits: regulation of TGF-β signalling (AAdis nominal $p$ value = $2.76 \times 10^{-5}$; FDR < 0.01), IGF binding (AAdis nominal $p$ value $1.47 \times 10^{-4}$; FDR < 0.01) and PDGF binding (AAdis nominal $p$ value $7.19 \times 10^{-5}$; FDR < 0.01). VEGF signalling was

significantly associated with ascending aortic distensibility (AAdis nominal $p$ value $5.15 \times 10^{-5}$; FDR < 0.01).

By averaging eQTL effect directions in aortic tissues, we can identify some trends in the directionality of these pathway associations. For example, we identify a number of eQTLs which suggest that increased expression of genes which enhance TGF-β signalling pathway (e.g., *WWP2*[36], *LRP1*[37]), or reduced expression of genes which inhibit TGF-β signalling (e.g., *THSD4*[38], *FGF9*[39]), may be associated with decreased distensibility and increased aortic areas (*WWP2* average NES −0.54 for AAdis, NES 0.33 for AAmin; *LRP1* NES 0.18 for DAmin; *THSD4* average NES 0.21 for AAdis, NES −0.22 for AAmin; *FGF9* NES 0.28 for AAdis, NES 0.28 for AAmin).

The gene with the greatest averaged NES for variants associated with AAdis is *GPX7* (average NES for AAdis 0.83), a gene encoding glutathione peroxidase 7, which is protective against oxidative stress[40] and has previously been associated with both ischaemic stroke and Parkinson's disease[41]. Notably, *SVIL*, a gene previously associated with ascending and descending aortic diameter[30], has a negative NES for AAdis, but a positive NES for DAmin, implying that increased expression of *SVIL* is associated with reduced distensibility and increased aortic dimensions. Also notably, eQTLs for *ESR1*, which are associated with aortic areas have a negative averaged NES− implying that increased *ESR1* expression is associated with smaller aortic areas.

For a full analysis of averaged NES, see Supplementary Fig. 13 and Supplementary Data 15c.

## Co-expression network analyses

Using expression data from single cell transcriptomics of the primate aorta[42], we generated co-expression modules for aortic endothelial and aortic smooth muscle cells. Using our MAGMA (adj. $p$ value < 0.01) and FUMA gene-based associations (see Methods), we generated functional sub-networks for each aortic trait, highly enriched for our significant genes and identified hub genes for modules expressed in aortic endothelial cells and aortic smooth muscle (see Fig. 3 and Supplementary Figs. 14–17). These hub genes include genes involved in smooth muscle cell contraction and differentiation (e.g., *ACTB, MYH10, MYL9, NEXN, ARIDSB* and *SVIL*), as well as others associated with TGF-β signalling. In endothelial cells, hub genes identified included *EDN1* and *GATA2*.

We used these co-expression modules for pathway analyses as described in Methods. The importance of extracellular matrix, vascular smooth muscle cell contraction and developmental pathways were highlighted by enrichment of GO biological pathway and molecular function terms (e.g. extracellular matrix organisation, collagen-containing extracellular matrix, contractile fibre, muscle contraction, actin binding, myosin binding, cardiovascular system development). GO terms related to the TGF-β pathway were also significantly enriched

**Table 3 | Most significant 30 genes associated with ascending and descending aortic distensibility (gene-based analysis using MAGMA v1.08 as implemented in FUMA v1.3.6)**

| AA distensibility | | | | | | DA distensibility | | | | | |
|---|---|---|---|---|---|---|---|---|---|---|---|
| CHR | START | STOP | ZSTAT | P (adj) | SYMBOL | CHR | START | STOP | ZSTAT | P (adj) | SYMBOL |
| 1 | 52870236 | 52886511 | 5.01 | 2.68E-07 | *PRPF38A* | 2 | 20883788 | 21022882 | 5.6592 | 7.61E-09 | *C2orf43* |
| 1 | 52873954 | 53019159 | 4.65 | 1.67E-06 | *ZCCHC11* | 3 | 186935942 | 187009810 | 6.3839 | 8.63E-11 | *MASP1* |
| 1 | 53068044 | 53074723 | 4.96 | 3.60E-07 | *GPX7* | 5 | 95726119 | 95769847 | 4.0163 | 2.96E-05 | *PCSK1* |
| 3 | 37027357 | 37034795 | 4.52 | 3.14E-06 | *EPM2AIP1* | 7 | 73442119 | 73484237 | 4.4229 | 4.87E-06 | *ELN* |
| 3 | 123798870 | 124445172 | 4.85 | 6.25E-07 | *KALRN* | 8 | 75512010 | 75735548 | 4.0596 | 2.46E-05 | *RP11-758M4.1* |
| 4 | 146678779 | 146859787 | 4.60 | 2.14E-06 | *ZNF827* | 8 | 75736772 | 75767264 | 4.3374 | 7.21E-06 | *PI15* |
| 5 | 81575281 | 81682796 | 4.68 | 1.40E-06 | *ATP6AP1L* | 8 | 92114060 | 92231464 | 5.0794 | 1.89E-07 | *LRRC69* |
| 5 | 95726119 | 95769847 | 6.11 | 5.00E-10 | *PCSK1* | 8 | 122624356 | 122653630 | 4.6601 | 1.58E-06 | *HAS2* |
| 5 | 122424816 | 122529960 | 5.10 | 1.69E-07 | *PRDM6* | 9 | 116638562 | 116818871 | 4.6059 | 2.05E-06 | *ZNF618* |
| 6 | 12290596 | 12297427 | 4.72 | 1.18E-06 | *EDN1* | 10 | 95753746 | 96092580 | 7.3635 | 8.96E-14 | *PLCE1* |
| 7 | 73442119 | 73484237 | 7.11 | 5.85E-13 | *ELN* | 10 | 96162261 | 96295687 | 5.8888 | 1.95E-09 | *TBC1D12* |
| 8 | 38585704 | 38710546 | 4.58 | 2.38E-06 | *TACC1* | 10 | 96305547 | 96373662 | 5.2578 | 7.29E-08 | *HELLS* |
| 8 | 75512010 | 75735548 | 5.26 | 7.03E-08 | *RP11-758M4.1* | 11 | 61535973 | 61560274 | 4.2122 | 1.26E-05 | *TMEM258* |
| 8 | 75736772 | 75767264 | 6.13 | 4.27E-10 | *PI15* | 12 | 33527173 | 33592754 | 4.8085 | 7.60E-07 | *SYT10* |
| 8 | 122624356 | 122653630 | 6.11 | 5.00E-10 | *HAS2* | 12 | 38710380 | 38717784 | 4.4827 | 3.69E-06 | *ALG10B* |
| 10 | 95753746 | 96092580 | 5.22 | 9.07E-08 | *PLCE1* | 12 | 57489191 | 57525922 | 4.3707 | 6.19E-06 | *STAT6* |
| 11 | 69924408 | 70035634 | 5.15 | 1.34E-07 | *ANO1* | 12 | 94071151 | 94288616 | 4.6778 | 1.45E-06 | *CRADD* |
| 12 | 21950335 | 22094336 | 5.58 | 1.19E-08 | *ABCC9* | 14 | 92335756 | 92414331 | 5.5917 | 1.12E-08 | *FBLN5* |
| 12 | 57489191 | 57525922 | 5.26 | 7.24E-08 | *STAT6* | 15 | 32737307 | 32747835 | 4.6319 | 1.81E-06 | *GOLGA8O* |
| 15 | 71389291 | 72075722 | 5.27 | 7.01E-08 | *THSD4* | 15 | 74218330 | 74244478 | 4.3266 | 7.57E-06 | *LOXL1* |
| 15 | 78916461 | 79020096 | 4.96 | 3.47E-07 | *CHRNB4* | 15 | 91426925 | 91439006 | 4.7615 | 9.61E-07 | *FES* |
| 16 | 75327596 | 75467383 | 4.83 | 6.82E-07 | *CFDP1* | 16 | 75327596 | 75467383 | 6.6103 | 1.92E-11 | *CFDP1* |
| 16 | 75446582 | 75498604 | 4.59 | 2.19E-06 | *RP11-77K12.1* | 16 | 75446582 | 75498604 | 6.2293 | 2.34E-10 | *RP11-77K12.1* |
| 16 | 88941266 | 89043612 | 5.06 | 2.12E-07 | *CBFA2T3* | 16 | 75476952 | 75499395 | 6.0227 | 8.58E-10 | *TMEM170A* |
| 16 | 89006197 | 89017932 | 5.07 | 1.98E-07 | *RP11−830F9.6* | 16 | 75510949 | 75529282 | 4.8918 | 4.99E-07 | *CHST6* |
| 17 | 1957448 | 1962981 | 5.39 | 3.45E-08 | *HIC1* | 16 | 89724210 | 89737680 | 4.1797 | 1.46E-05 | *SPATA33* |
| 17 | 1963133 | 2207065 | 8.24 | 8.86E-17 | *SMG6* | 17 | 1963133 | 2207065 | 6.0905 | 5.63E-10 | *SMG6* |
| 17 | 2206677 | 2228554 | 7.68 | 7.83E-15 | *SRR* | 17 | 2206677 | 2228554 | 5.3263 | 5.01E-08 | *SRR* |
| 17 | 2225797 | 2240801 | 5.33 | 4.80E-08 | *TSR1* | 19 | 39138289 | 39222223 | 5.3381 | 4.70E-08 | *ACTN4* |
| 20 | 47240790 | 47444420 | 4.61 | 2.04E-06 | *PREX1* | 19 | 39220827 | 39260544 | 4.6378 | 1.76E-06 | *CAPN12* |

*CHR* chromosome, *ZSTAT* z statistic from MAGMA[35], P*(adj)* adjusted p values using Bonferroni correction for 19,088 protein-coding genes.

in functional modules derived from gene associations with all the aortic phenotypes (FDR < 0.05) (see Supplementary Data 16 for full results).

**Phenome-wide association studies and MR-PheWAS**
We performed a Phenome-Wide Association Study (PheWAS) and a subsequent MR−Phenome-Wide Association Study (MR-PheWAS) to explore associations between our aortic traits of interest and clinical diagnoses in the whole UK Biobank population for which relevant clinical data were available (*n* = up to 406,827), controlling for age, sex, and for genotype array and four principal components of genotype for the MR-PheWAS analysis.

Initial PheWAS identified significant phenotypic associations between aortic traits and multiple hypertension-related clinical codes. Aortic areas showed a positive phenotypic association with hypertension (for example, AAmin $\beta$ = 0.001; $p$ = 2.5 × 10$^{-21}$; see Supplementary Data 17 for full results). Aortic distensibility was negatively associated with hypertension ($\beta$ = −0.217; $p$ = 1.65 × 10$^{-25}$). There was a significant negative association for all traits with type II diabetes mellitus.

A subsequent analysis of genotypic associations using MR-PheWAS supported a significant causal relationship between AAdis (MTAG) and aortic aneurysms (Inverse Variance Weighted (IVW) OR = 0.28, 95% CI 0.16–0.50; $p$ value 2.14 × 10$^{-5}$; consistent directions

of effect with Weighted Median (WM)/MR-Egger and with use of the single-trait analysis of AAdis as the genetic instrument, though the latter did not reach significance), suggesting clinical meaningfulness of the distensibility phenotype. MR-PheWAS also suggested that ascending and descending aortic areas are causally related to the risk of aortic aneurysms without evidence of significant pleiotropy (see Supplementary Data 18 for full results).

**Relationship between blood pressure and aortic dimensions**
We tested further for bidirectional causal relationships between quantitative blood pressure traits (using GWAS summary statistics from a previous study[43]) and aortic areas using MR. MR results supported a bidirectional causal relationship between ascending aortic areas and diastolic blood pressure (DBP; AAmin->DBP; $\beta_{IVW}$ = 0.004, $p$ = 4.3 × 10$^{-16}$; DBP-> AAmin: $\beta_{IVW}$ = 6.6; $p$ = 2.5 × 10$^{-5}$) and between ascending aortic area and pulse pressure (PP; AAmin->PP; $\beta_{IVW}$ = −0.007; $p$ = 7.1 × 10$^{-15}$; PP-> AAmin: $\beta_{IVW}$ = −8.1, $p$ = 4.4 × 10$^{-7}$). MR-Egger estimates were consistent for all but the DBP-> AAmin analysis, for which the estimates were in the opposite direction (Supplementary Data 19a−c). Contamination mixture MR (MR-ConMix, see Methods) showed consistent findings for all analyses (Supplementary Data 20). Similar analyses for causal relationships with blood pressure were not performed for distensibility since blood pressure is used for calculation of the trait.

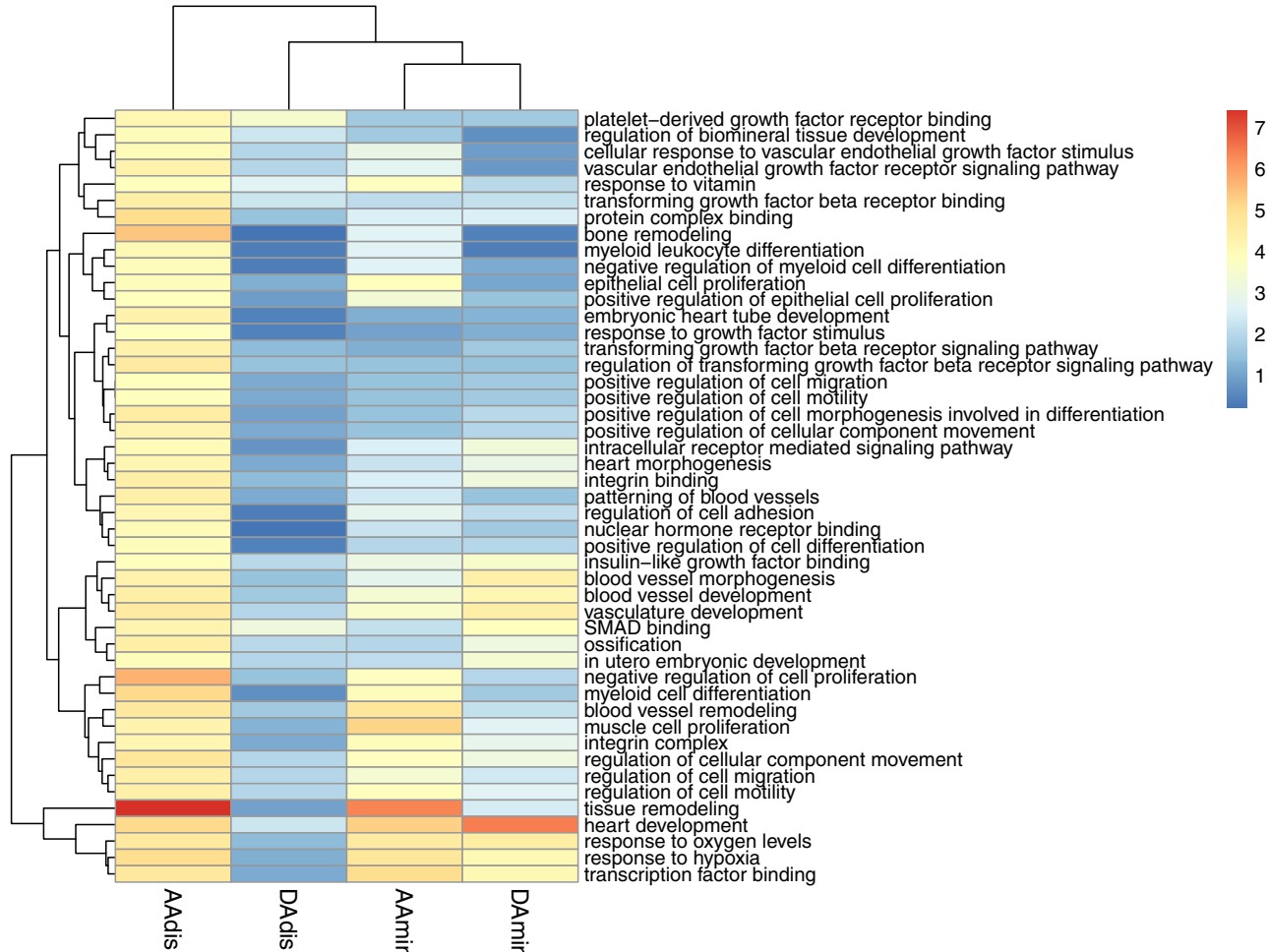

**Fig. 2 | Heatmap of significantly enriched gene ontologies (GO terms) for minimum area and distensibility phenotypes generated by DEPICT.** Colour scale denotes the significance of enrichment, (unadjusted $p$ value −log10 scale). Only GO terms significantly enriched in association with AAdis (FDR < 0.01) are presented. Full results can be found in Supplementary Data 11c. AAdis ascending aortic distensibility, AAmin ascending aortic minimum area, DAdis descending aortic distensibility, DAmin descending aortic minimum area.

## Genetic relationships between aortic traits and cerebral small vessel disease or cervical artery dissection

We explored genetic correlations and potential causal relationships between aortic traits and brain SVD estimated from the brain MRI measure of WMH burden in 50,970 individuals[44]. Using LDSC, we identified a significant genetic overlap between all aortic traits and WMH burden which defined a positive association with minimum aortic area and an inverse association for the distensibility measures (AAmin $r_g$ = 0.20, $p$ = 0.001; DAmin $r_g$ = 0.22, $p$ = 2.19 × 10⁻⁵; AAdis $r_g$ = −0.22, $p$ = 5.0 × 10⁻⁴; DAdis $r_g$ = −0.33; $p$ = 1.0 × 10⁻⁴, see Supplementary Data 21). In further analyses we found no significant genome-wide overlap between aortic traits and risks of cervical artery dissection, a leading cause of stroke in young people which was associated with aortic phenotypes in an earlier study[45]. However, the regional level overlap estimates from a Bayesian pairwise GWAS (GWAS-PW) suggest a high probability of shared variants between aortic distensibility traits and the cervical artery dissection (CeAD) risk locus *PHACTR1-EDN1* (see Supplementary Data 22). Most of the other CeAD risk loci with high probability of shared variants with aortic traits harbour single nucleotide variants (SNVs) associated at genome-wide significance with one or more of the aortic traits. Exceptions to this include region at chromosomes 12 (including *c12orf49, RNFT2, PAWR, OTOGL*), 16 (including *CMIP, PKD1L2, BCO1*) and 2 (including *MBD5, EPC2, LYPD6B*), suggesting further novel biologically relevant associations with aortic traits in these regions.

## Exploring relationships between aortic traits and white matter hyperintensity burden using MR

A lower aortic distensibility or a greater ascending aortic area is genetically correlated with an increased burden of WMH in our data. We hypothesised that there might be a causal relationship between these aortic traits and cerebral small vessel disease. Although a two-sample MR using genetic associations with aortic traits as the instrumental variable and WMH as the outcome showed no evidence for a causal association for any of the aortic traits after multiple testing corrections (see Supplementary Data 23 for results), after accounting for the effect of blood pressure (either systolic or pulse pressure) using a multivariable MR, we found evidence for a direct causal effect of both ascending and descending aortic distensibility and ascending aortic area on WMH burden. Lower distensibility and higher area were associated with an increased WMH burden (accounting for systolic blood pressure, for AAdis $\beta$ = −0.12, $p$ = 1.49 × 10⁻³ and DAdis $\beta$ = −0.21, $p$ = 1.14 × 10⁻³ using MTAG-derived associations as the instrumental variable and for AAmax $\beta$ = 4.0 × 10⁻⁴, $p$ = 1.26 × 10⁻³ and AAmin $\beta$ = 3.8 × 10⁻⁴, $p$ = 2.91 × 10⁻³ using stage 1 associations as the instrumental variable; see Supplementary Data 24 for full results). We attempted to replicate this association directly using multivariable two-sample MR (MVMR) in a smaller cohort ($N$ = 3317) from the Rhineland study, for whom WMH data and genotyping data were available. In this underpowered analysis, compounded by low conditional F statistics ($F$ < 10), we were not able to demonstrate consistent

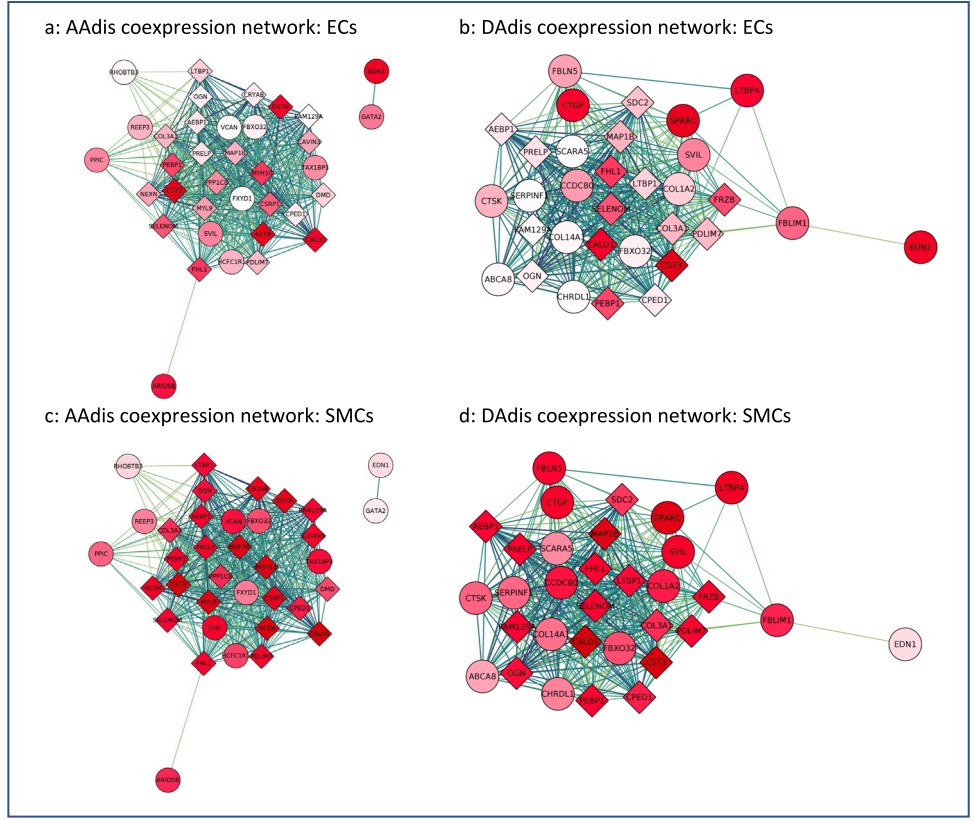

**Fig. 3 | Co-expression networks for aortic distensibility GWAS genes generated with primate single cell expression data for the aorta[42].** The co-expression networks were derived from extended models ($r > 0.2$) in aortic endothelial (ECs) and aortic smooth muscle cells (SMCs). Round circles represent genes which were significantly associated (unadjusted $p$ value $<5 \times 10^{-8}$) with an aortic trait in the current GWAS. Diamonds represent other genes significantly co-expressed in the published single-cell data for the cell-type indicated. The deeper the shade of red, the higher the level of expression of that gene in the specified cell-type. The strength of co-expression is denoted by the colour of the lines joining genes with higher correlations indicated by darker lines. "Hub genes" are found in the centres of these modules. **a** Co-expression networks derived from genes associated with ascending aortic distensibility (AAdis) and expression data from aortic endothelial cells (ECs). **b** Co-expression networks derived from genes associated with descending aortic distensibility (DAdis) and expression data from aortic endothelial cells (ECs). **c** Co-expression networks derived from genes associated with AAdis and expression data from aortic smooth muscle cells (SMCs). **d** Co-expression networks derived from genes associated with DAdis and expression data from aortic smooth muscle cells (SMCs). See Supplementary Figs. 14–17 for further co-expression results.

replication, although the direction of effect was consistent for the distensibility traits. Results of the analysis are shown in Supplementary Data 25.

## Discussion

Our analysis provides a first large-scale GWAS of ascending and descending aortic distensibilities, and adds substantively to the literature concerning the genetic basis of variation in aortic dimensions. We show that aortic distensibility has a significant heritable component, with 11% of the variance in AA distensibility and 10% of the variance in DA distensibility explained by the common genetic variants included in our study (increasing to 24% and 21% respectively for the MTAG analysis). We identify 38 loci associated with these measures of aortic stiffness, and a total of 31 novel loci for aortic traits including aortic areas (102 loci overall). Annotation of these loci provides evidence for mechanistic associations of TGF-β, IGF, PDGF and VEGF signalling pathways with aortic distensibility.

The clinical significance of our findings is suggested by the potential causal associations between AAdis (and other aortic traits) and aortic aneurysms defined by MR-PheWAS. Multivariable MR provides new evidence for possible mechanistic associations between cerebral small vessel disease and both aortic distensibilities and aortic area, helping to explain their long-recognised clinical relationships[20,44].

Mendelian aortopathy or cardiovascular disease-associated genes *ELN*[46], *THSD4*[38], *FBLN5*[47,48], *PRDM6*[49] and *ABCC9*[50] directly overlap loci

associated with aortic distensibility phenotypes and thus are strong candidates for expression of functional effects of variation at the corresponding loci. Similarly, Mendelian disease genes *FBN1*[51], *MYH7*[52], *TBX20*[53], *MASP1*[54] and *LOX*[55] overlap loci associated with aortic area phenotypes. Other genes associated with aortic area in our analyses have previously been associated with risks of acute aortic dissection (*ULK4, LRP1*[56]). Several gene ontologies were associated with our measured aortic traits which are also of significance in Mendelian aortic disease[57]. Those GO terms related to the extracellular matrix, cardiovascular development and vascular smooth muscle cell function (with genes such as *ELN, ABCC9, ANO1* and *PRDM6* associated with AAdis) were amongst the most consistently identified. This overlapping genetic landscape of distensibility (and aortic areas) and aortic aneurysms, and our finding of a likely causal link between these phenotypes using MR-PheWAS suggest that functional pathways related to genes associated with quantitative aortic traits contribute to the pathogenesis of the aortic disease. Together, these observations also support the growing consensus that cardiovascular disease phenotypes may be expressed as a result of extremes of normal genetic variation in the population[58], and support the use of distensibility to predict both aneurysm formation and progression[4].

Our data also support the role for TGF-β signalling in determining aortic distensibility in the general population. The TGF-β pathway has long been recognised as an important modulator of aortic function, with variants in many of the major components identified as causal in

Mendelian aortic disease such as Loeys-Dietz syndrome[59]. Our data highlight less well-known TGF-β pathway components such as *LTBP4* and *HIPK3* ($Z$ score = 4.33, $p = 7.5 \times 10^{-6}$) through their association with ascending aortic distensibility. Our eQTL analysis suggests that variants associated with increased expression of TGF-β pathway components are also associated with decreased distensibility, and variants associated with increased expression of TGF-β pathway inhibitors are associated with increased distensibility. This is in keeping with previous observations that TGF-β signalling is up-regulated in aortic diseases[60–62], and that aortic distensibility is reduced in several of these conditions[2,4]. Gene ontologies and cell-specific co-expression modules associated with the measured aortic traits suggest that IGF, PDGF and VEGF signalling also play significant roles in determining aortic biology. IGF signalling plays a fundamental role in development and tissue homoeostasis[63] and may modulate smooth muscle cell turnover and affect smooth muscle cell phenotype[64]. The roles of insulin and of insulin-like growth factor signalling are of considerable therapeutic interest for aortic pathology given recent evidence suggesting that metformin, a known regulator of both signalling pathways[65], could be an effective treatment for abdominal aortic aneurysm, and the consequent initiation of clinical trials testing this[66,67].

Each of these gene sets offers interesting candidates in the search for new Mendelian aortic disease genes. Our results also add to the literature on sex differences in the genomic regulation of cardiovascular traits[68,69], with new evidence presented here suggesting distinct, biologically relevant associations in males and females and implicating genes such as *ADAMTS7*, *SCAI*, *ALDH2* and *FES* as sex-specific determinants of aortic traits and thus possibly also the related diseases.

The strongest SNV association with ascending aortic distensibility was found in close proximity and upstream of *ELN*, the gene encoding elastin, a functionally central component of aortic elastic fibres. Many aortic pathologies – both genetic and acquired – share a final common pathway of degradation of elastic fibres and elastin in the aortic media. It is notable that elastin production occurs at very low levels in adults (it is almost undetectable in mice >3 weeks old)[70]. Whether the association with aortic distensibility arises developmentally through impact of the SNP on elastin expression or its timing, through impact on elastic fibre structure, or via an effect in later life on signalling via elastin breakdown products[71], remains to be elucidated. Insoluble elastin also functions to regulate vascular smooth muscle cell proliferation, with a functional haploinsufficiency of *ELN* causing the Mendelian disease Williams-Beuren syndrome, which is characterised by supravalvular aortic stenosis amongst other systemic features[46,72]. *FBLN5* was strongly associated with DAdis. This encodes fibulin-5, a secreted, extracellular matrix protein and a mediator of elastic fibre assembly[73]. Variants in *FBLN5* cause a Mendelian form of *cutis laxa* associated with aortic aneurysm, vascular tortuosity and supravalvular aortic stenosis[47,48]. The importance of extracellular matrix (ECM) composition and regulation is demonstrated by the identification of multiple ECM-related GO terms associated with aortic phenotypes. Specific gene associations also serve to emphasise this, including three members of the ADAMTS family, which regulate ECM turnover: *ADAMTS7* and *ADAMTS8*, both previously associated with aortic minimum areas and replicated here, and a novel association of ascending aortic distensibility with *ADAMTS9*. The strong association of AAdis with *HAS2* (encoding a hyaluronan synthetase) demonstrated that glycosaminoglycan components of the ECM are also key determinants of aortic traits.

Other specific associations provide insights into the complexity of aortic biology. For example, the second most significant association with AAdis is within a long, non-coding RNA (lncRNA) just 3′ of *PCSK1*, a proprotein convertase whose substrates include many hormones such as renin, insulin and somatostatin (associated previously with body mass index and obesity[74]) and therefore which may mediate multiple endocrine influences on aortic traits. The third most significant locus associated with AAdis was previously associated with coronary artery

disease[75] and spans *SMG6* (a regulator of nonsense-mediated decay) and *SRR*, a serine racemase. The causal gene at this locus is thought to be *SMG6*, although functional data demonstrating strong eQTLs for *SRR* in all the risk alleles identified suggests it remains a candidate gene for this locus.

The shared genetic basis of ascending and descending distensibilities is limited (Supplementary Fig. 4), consistent with the different developmental origins of these parts of the aorta, and associated differences in structures of the aortic wall, in which elastin content diminishes more distally[76]. The most significant association for descending aortic distensibility is in *PLCE1*, a gene previously associated with blood pressure traits[43,77]. Evidence from knockout mice suggests that PLCE1 contributes to the integration of β-adrenergic signalling with inputs from IGF-1 and other pathways to regulate cardiomyocyte differentiation and growth. We speculate that it might play a similar integrative role in the development and remodelling of aortic tissues.

Associations between aortic traits and brain small vessel disease have long been recognised, but the mechanisms responsible have not been defined clearly[20,78]. This has been a particularly difficult relationship to untangle, as both are subject to confounding influences of blood pressure and other pleotropic factors. Our multivariate MR provides novel evidence suggesting that aortic traits including distensibility are causally linked to WMH and that this relationship is independent of (and additive to) the effects of blood pressure. By inference, as WMH burden predicts cognitive decline and dementia (with evidence supporting a causal association with Alzheimer's type dementia[16,44]), these results indirectly suggest that aortic stiffening could also contribute to cognitive decline and dementia, e.g. through altered haemodynamics and resultant changes in cerebral blood flow leading to effects on brain endothelial cell function and small vessel remodelling[79,80].

A shared genomic influence on aortic distensibility and cervical artery dissections was identified at the *PHACTR1/EDN1* locus. Previously, this locus was implicated in coronary artery disease[81], myocardial infarction[82], migraine[83], fibromuscular dysplasia[84] and cervical artery dissection[85]. We demonstrated an association of *EDN1* with ascending aortic distensibility, and further characterised *EDN1* as a hub gene in co-expression networks derived from aortic endothelia, suggesting that *EDN1* may be responsible for (or functionally contribute to) this shared genetic association with both CeAD risk and aortic distensibility,

Although we have made several novel observations, there are obvious limitations of our study. The GWAS was restricted to the analysis of Caucasian individuals and additionally, it is well-recognised that UK Biobank is not representative of the UK population as a whole[86]. While genotype-phenotype associations can be biased by population stratification, our analysis was adjusted for ethnicity and relatedness. Nonetheless, the "healthy volunteer" selection bias of the cohort could be a potential confound if it significantly influenced the aortic traits of interest[87]. To address this potential confound we now have demonstrated replication in an independent European population cohort (SHIP) for our primary GWAS findings and have shown that the findings in UK Biobank can be used to predict related cardiovascular traits (WMH) in this independent cohort. Second, the accuracy (and possibly also the precision) of the distensibility measures likely was reduced by the need to use non-invasive blood pressure measurements (acquired on the same day as the imaging) as proxies for central blood pressure recordings. The ascending and descending aortic distensibility measures also suffer from confounding due to a likely bidirectional relationship with blood pressure, given the use of blood pressure indices in the derivation of the phenotype. Uncontrolled confounding from residual effects of blood pressure could bias the MR analyses. Nevertheless, our genetic associations were significantly enriched for genes expressed in the aorta and identified

genes known to be important in aortic biology, affording some confidence in the robustness of our results.

In summary, our results provide genetic association data highlighting roles for TGF-β and other growth factors (IGF, PDGF, VEGF) signalling pathways in the elastic function of the aorta and, by inference, in aortic disease. We present new evidence for potential causal links between lower aortic distensibility and increased risk of aortic aneurysm and for common causal mechanisms relating cerebral small vessel disease and aortic structure and function that could explain the clinically observed relationships between late-life cognitive decline and aortic disease[14,18]. A better understanding of the underlying mechanisms based on these genetic data could lead to the identification of new therapeutic targets for the reduction of both cardiovascular disease and dementia risks.

## Methods

### Data: main cohort

The UK Biobank CMR imaging was conducted using a rigorously controlled acquisition protocol[88]. The cohort has been well-described previously and ethics approval for UK Biobank has been given by the North West Multi-Centre Research Ethics Committee (MREC). These analyses were conducted under Application number 18545. The mean age at the time of CMR was 64 ± 8 years (range 45–82, 49% of participants were male [see Supplementary Data 1]). Exclusion criteria for imaging included a range of relative contraindications to magnetic resonance imaging scanning as well as childhood-onset disease and pregnancy. Aortic cine images were acquired using transverse bSSFP sequence at the level of the pulmonary trunk and right pulmonary artery on clinical wide bore 1.5 Tesla scanners (MAGNETOM Aera, Syngo Platform VD13A, Siemens Healthcare). Each cine image sequence consists of 100 time frames. The typical image size is 240 × 196 pixel with the spatial resolution of 1.6 × 1.6 $mm^2$. Brachial blood pressure was obtained using a manual sphygmomanometer and converted into central blood pressure for the distensibility calculations by applying a brachial-to-aortic transfer function using the Vicorder software[88].

### Derivation of imaging phenotypes: main cohort

A recurrent convolutional neural network was developed for aortic image segmentation and trained using manual annotations of 800 ascending aorta and descending aorta images (400 subjects and 2 time frames per subject)[25,27] The network was applied to segmenting aortic images across the cardiac cycle. A semi-automated quality control was performed for all segmentations, consisting of automated checking of missing or fragmented segmentation and subsequent manual checking on segmentation screenshots. Quality control (QC) used the following criteria: (1) the aorta appears in all the time frames of the image sequence; (2) there is no abrupt change of aortic areas between adjacent time frames; (3) the aortic segmentation constitutes a single connected component. Any segmentations that did not fulfil these criteria were excluded from analyses. For validation of the automated segmentation, an image analyst trained for cardiac imaging visually segmented images. By comparing automated and manually segmented images, we found that the neural network achieves a Dice metric of 0.960 for the ascending aorta and 0.953 for the descending aorta as reported previously[25]. The Dice metric is a commonly used metric for evaluating image segmentation accuracy. A Dice metric of over 0.95 is typically regarded as of high accuracy.

Six aortic imaging phenotypes were calculated based on the automated segmentations, including those for the maximal area, minimal area and distensibility for both the ascending aorta and descending aorta[4]. Distensibility was calculated as

$$Dis = \frac{A\max - A\min}{A\min \times (SBP - DBP)} \quad (1)$$

where Amax and Amin denote the maximal and minimal area and SBP and DBP denote the systolic and diastolic central blood pressure, measured at the imaging visit during the study protocol (measured brachially and converted to central blood pressure by applying a brachial-to-central transfer function as described above). Aortic images were available for 37,891 subjects. After running the image segmentation pipeline and performing quality control, imaging phenotypes were available for 36,995 participants.

### Genomic analyses: main cohort

We performed stage 1 GWAS on six imaging phenotypes (AAmax, AAmin, DAmax, DAmin, AAdis and DAdis). Outliers with phenotype values >4 standard deviations (SDs) from the mean were excluded to ensure we did not include patients with undiagnosed aortic aneurysm in our results. After exclusions for image quality control, outlying BMI (< 15 or >40), stage IV hypertension, missing covariates, diagnosis of aortic disease and non-white ethnicity, 4,405 participants were excluded leaving 32,590 individuals for the GWAS of AAmax, AAmin, DAmax, and DAmin, and 29,895 for the GWAS of AAdis and DAdis (the latter figure is lower due to more missing contemporaneous blood pressure recording data and incomplete imaging sets). See Supplementary Fig. 1 for more details on exclusions. After exclusions, we rank-normalised the distensibility phenotypes due to the non-normal distribution of the distensibility phenotypes (Supplementary Fig. 2). Aortic area phenotypes approximated a normal distribution and so raw areas were used to facilitate interpretation of the effect sizes. The genetic model was adjusted for age at the time of imaging, sex, mean arterial pressure, height, and weight. We used the linear mixed model approach implemented in BOLT-LMM (v2.3.4)[89]. The genetic relationship matrix (GRM) constructed by BOLT was based on all directly genotyped SNPs ($N = 340,336$) passing the threshold settings (MAF > 0.05, $p$(HWE) > $1e^{-6}$ and genotype calling rate >98.5). For the main analysis, a threshold of MAF > 0.01 was applied to the SNPs. Genomic inflation (lambda) was calculated in R as the median chi-square values derived from the $p$ values divided by the expected median of a chi square distribution with 1 degree of freedom. Power heatmaps are provided in Supplementary Fig. 18 for a range of MAF and standardised betas appropriate for our cohort sizes (whole cohort and sex-specific). Calculation using gwas-power (v1) in R (which uses the formulae in Appendix 1 of Visscher et al.[90]) and checked with Quanto (https://pphs.usc.edu/download-quanto/) suggest that the power with our full cohort size of 32,590 for a beta of 0.05 at a MAF of 0.3 is 0.66 to detect genome-wide significant associations at $p < 5 \times 10^{-8}$. Increasing the sample size by 10,000 individuals would increase the power to detect a beta of 0.05 at a MAF of 0.3, to 0.89. These figures motivated us to increase power using MTAG as described.

We used MTAG (version 1.0.8)[28] for multi-trait analysis of GWAS summary statistics to increase power. MTAG can identify genetic loci associated with a particular phenotype where the single-trait GWAS is underpowered. The method uses the correlation structure of the trait in question with other traits to boost power. MTAG has been described in detail in Turley et al[28] and used across many published GWAS studies (from our group and many others) to boost power[58,91,92]. The key idea is that when there is a correlation between GWAS estimates from different traits, it is possible to improve the accuracy and power of the effect estimates for each individual trait by including information contained in the GWAS estimates for the other traits. It implements a generalisation of inverse variance-weighted meta-analysis, assuming a constant variance:covariance matrix of effect sizes across traits. It therefore may fail to identify trait-specific loci but will increase the power to detect loci associated with the other related and correlated traits. We used all 6 aortic phenotypes for our MTAG analysis. Regression coefficients (beta) and their standard errors were used for MTAG. The results of the multi-trait analyses are shown in Fig. 1c, d and in Supplementary Figs. 7, 8, as well as Supplementary Data 6.

We additionally conducted a sex-specific analysis by performing GWAS on aortic areas for autosomal SNPs in men and women separately, using the BOLT-LMM pipeline as above, and compared the sizes of the sex-specific genetic associations for SNPs with a $P$ value smaller than $5 \times 10^{-8}$ using a z test[93]. We did not repeat this analysis for distensibility phenotypes as it was underpowered because of the reduced cohort size and smaller effect sizes for the distensibility SNPs as well as lower heritability estimates for these traits.

To classify genomic loci associated with our imaging phenotypes, GWAS summary statistics were processed using FUMA (v1.3.6)[34] and a pre-calculated LD structure based on the European population of the 1000 Genome Project[94]. SNPs that reached genome-wide significance ($p < 5 \times 10^{-8}$) and with $r^2 < 0.6$ were defined as independently significant. All variants with $r^2 \geq 0.6$ were labelled as candidate variants for further annotation by FUMA. In a second clumping procedure to define lead SNPS, those correlated with $r^2 < 0.1$ were defined as independently significant. Finally, proximal LD blocks of independent significant SNPs with <250 kb distance were merged and considered as a single genomic locus. To consolidate genomic loci across different traits, the merge function implemented in bedtools (v2.29.2) was applied[95].

For the association of SNPs with genes, biological processes and tissue expression, we applied the SNP2GENE function implemented in FUMA to the summary SNP statistics. For the positional mapping of SNPs to genes, a maximal distance of 10kB was set. eQTL mapping was performed based on the aorta tissue samples in GTEx v8 (using only gene pairs with significant SNPs with the default settings of FDR < 0.05 or $p$ value <1e$^{-3}$). 3D chromatin interaction mapping was based on HiC aorta data (GSE87112) within a promotor region window of 250 bp upstream and 500 bp downstream from the transcriptional start site and a threshold for significant loops of FDR < 1e$^{-6}$. Enhancer and promotor regions were annotated using the aorta epigenome (E065) from the Roadmap Epigenome Project (http://www.roadmapepigenomics. org/). MAGMA (Multi-marker Analysis of GenoMic Annotation) was employed to obtain the significance of individual genes with the specificity of tissue expression based on 54 types in GTEx v8 and the association with 10,678 gene sets from MsigDB v6.2 (with 4761 curated gene sets and 5917 GO categories). Further annotation of significant SNPs and loci was performed manually with SNP lookups in GTEx v8[29]; normalised effect sizes (NES) are reported from this data set using "Artery-Aorta" for the main analysis and including other arterial tissue types ("Artery – Tibial" and "Artery – Coronary") where stated.

To aggregate eQTL data over genes to investigate the directionality of associations between gene expression and imaging phenotypes, we aligned the direction of NES of eQTLs to the direction of the phenotype effect sizes (betas) obtained from BOLT or MTAG. Thus, in the "direction-corrected" data, a positive NES corresponds to the case where the effect allele has the same direction of association as the eQTL – ie the allele which is associated with increased phenotype value is also associated with increased gene expression while negative NES indicates that allele associated with increased phenotype value is associated with decreased gene expression. To facilitate interpretation, direction-corrected NES of eQTLs linked to a gene were subsequently aggregated by averaging. In total, 8326 eQTLs mapped by FUMA and linked to 164 unique candidate genes could be aligned to at least one of the six imaging phenotypes (presented in Supplementary Data 15c). Candidate genes associated with TGF, IGF, VEGF, PDGF pathways or ECM were selected based on KEGG and GO annotations.

For Gene Ontology (GO) enrichment analysis, the Data-driven Expression Prioritised Integration for Complex Traits (DEPICT) software was applied (v1 beta rel194, www.broadinstitute.org/depict), which is based on probabilistic memberships of genes across reconstituted gene sets[96] For LD-based clumping by PLINK (v2.0)[97], which precedes the DEPICT analysis, a $p$ value threshold of $10^{-5}$, a distance threshold of 500 kb and a LD threshold of 0.1 was set (following the recommendations on the DEPICT website). Note that DEPICT excludes any SNPs in the HLA region, on a sex chromosome, or not found in the 1000 Genomes Project data.

Comparisons between our data and those reported by Pirruccello et al.[30] and Tcheandjieu et al.[31] were made using the lead SNPs and corresponding beta-values. The overlaps are reported in Supplementary Data 11. To assess the degree of convergence of their studies with our results, lead SNPs were assigned to a genomic locus found in our study if they were within the locus or a 250 kb distance interval.

## Replication data: Study of Health in Pomerania (SHIP) cohort

Cohort characteristics and recruitment criteria have previously been reported[33] and details of the study population, image acquisition protocols, genotyping and statistical analysis are reported in *Supplementary Information*. Ethics approval was given by the medical ethics committee of the University of Greifswald. In brief, 2787 individuals recruited as part of two separate cohorts within this population study (SHIP and SHIP-Trend; see Supplementary Information for details), had baseline whole-body CMR scans. Aortic areas were manually measured from the axial images at the level of the pulmonary bifurcation. Genotyping was performed using Affymetrix Genome-Wide Human SNP Array 6.0 and Illumina Human Omni 2.5 array and imputation was performed in the Michigan Imputation Server using the HRC v 1.1 reference panel. Genome-wide linear regression analyses were performed in each cohort separately using EPACTS-3.2.9 (https://github. com/statgen/EPACTS) adjusted for sex, age, mean arterial pressure, body height, body weight, array type (SHIP-Trend only), and the first two genetic principal components. The results of both cohorts were subsequently meta-analysed using an inverse variance weighted method implemented in METAL (v2011-03-25)[98]. Lookups were performed in the meta-analysed data.

## LD Score regression

We performed LD Score regression using LDSC (LD SCore) v1.0.1 (https://github.com/bulik/ldsc)[94,99] to assess the heritability of the imaging phenotypes. The genetic correlation between imaging phenotypes and blood pressure by LD score was computed using 1000 Genomes European data[94]. We used the GWAS summary statistics for SBP, DBP and pulse pressure (PP) from the International Consortium for Blood Pressure (ICBP)[43] for the corresponding analyses for genetic correlation with blood pressure.

## Co-expression analysis

Co-expression networks for aortic phenotypes were derived using a single cell RNA-seq (scRNA-seq) data set for primate arteries[42]. The data were obtained from Gene Expression Omnibus (accession number GSE117715) and included read counts for over 9000 single cells from aortas and coronary arteries of 16 *Macaca fascicularis*. Low abundance genes were removed if they had read counts of less than 5% of the cells, leaving a total of 9903 genes for further analysis. The Bioconductor package *scater* (v1.14.6) was applied in R (v3.6.0) to compute log-transformed normalised expression values from the count matrix[100]. Subsequently, correlation of expression and its significance was derived using the *correlatePairs* function of the Bioconductor *scran* (v1.14.6) package, which calculated modified Spearman correlation coefficients and derived their significance using a permutation approach[101]. A basal co-expression network was constructed using gene pairs with significant correlation (false discovery rate; FDR < 0.01) and a minimum Spearman correlation coefficient *rho* of 0.1 or 0.2. Subsequently, sub-networks for imaging phenotypes were derived by retrieving gene pairs with at least one gene associated by MAGMA or FUMA with the specific phenotype.

Functional enrichment of genes connected with variants identified on GWAS was carried out using overrepresentation enrichment

analysis implemented in the Bioconductor *clusterProfiler* (v3.14.3) package[102]. The background gene set (or universe) was defined by the genes covered by the scRNA-seq data after exclusion of low abundance genes.

## Mendelian randomisation

To investigate potentially causal relationships between aortic phenotypes and diseases, we performed bidirectional two-sample MR using our GWAS for aortic imaging phenotypes and the International Consortium for Blood Pressure GWAS on blood pressure[43]. We limited our MR analyses to AAmax, AAmin, DAmax, and DAmin because blood pressure is included in the calculation of the distensibility measure.

For either direction of potential causal relationships between the aortic phenotypes and blood pressure, we selected SNPs associated with the exposure at genome-wide significance level ($P$ value $< 5 \times 10^{-8}$ and F-statistic > 10). For SNPs correlated with $r^2 > 0.1$, we used only the SNP with the smallest $p$ value for the SNP-exposure association. We tested the validity of the genetic variants as instrumental variables using the contamination mixture method (MR-ConMix) using the R package *MendelianRandomization* (v0.6.0)[103]. The contamination mixture method constructs a likelihood function based on the SNP-specific estimates and evaluates the SNP-specific contribution to the likelihood.

In each case, we estimated SNP-specific associations as the ratio of SNP outcome to exposure associations (Wald ratio)[104]. SNP-specific associations were combined using the inverse variance weighted (IVW) estimator[105]. The putative causal effect ($\beta_{IVW}$) of exposure on a given outcome was estimated using the inverse variance weighting (IVW) method as the weighted sum of the ratios of beta-coefficients from the SNP outcome associations for each variant (j) over corresponding beta-coefficients from the SNP-exposure associations ($\beta_j$). The ratio estimates from each genetic variant were averaged after weighting on the inverse variance ($W_j$) of $\beta_j$ across L uncorrelated SNPs,

$$\beta_{IVW} = \frac{\sum_{j=1}^{L} Wj\beta_j}{\sum_{j=1}^{L} Wj} \tag{2}$$

We also used weighted median (WM) and MR-Egger regressions as sensitivity methods to test the robustness of associations[105]. Potential horizontal pleiotropic effects were investigated using MR-Egger[106]. Outlier SNPs identified by MR-PRESSO were excluded from the analyses[107]. In an additional analysis, we tested the validity of the genetic variants as instrumental variables using MR-ConMix[103]. We accounted for multiple comparisons of four aortic imaging phenotypes, three blood pressure traits, and two directions using Bonferroni correction with a $P$ value threshold of 0.05/(4*3*2)=0.002.

MR analysis was also used to investigate the potential causal relationships between different aortic traits with WMH. In addition to IVW, WM, and MR-Egger, we implemented R package RadialMR (v1.0, available through CRAN repositories)[108]. A $p$ value < 0.01 correcting for 6 tests (for the 6 aortic traits) was considered significant. In the presence of heterogeneity ($P_{Het} < 0.01$, Cochran's Q statistic) due to horizontal pleiotropy, RadialMR was used in the identification of pleiotropic SNPs that have the largest contribution to the global Cochran's Q statistic by regressing the predicted causal estimate against the inverse variance weights. After excluding influential outlier SNPs, the IVW test was repeated along with MR-Egger regression in which the regression model contains the intercept term representing any residual pleiotropic effect[109]. Non-significant MR-Egger intercept was used as an indicator to formally rule out horizontal pleiotropy. Relative goodness of fit of the MR-Egger effect estimates over the IVW approach was quantified using $Q_R$ statistics, which is the ratio of the statistical heterogeneity around the MR-Egger fitted slope divided by the statistical heterogeneity around the IVW slope. A $Q_R$ close to 1

indicates that MR-Egger is not a better fit to the data and therefore offers no benefit over IVW[108].

## Multivariable MR

We also conducted MVMR using the MVMR package in R[110] to estimate the direct effect of aortic traits on the cerebral small vessel disease (cSVD) outcome (WMH) after accounting for potential confounding with blood pressure traits, by conditioning on every other explanatory variable included in the model. Different combinations of explanatory variables were considered and the $F_{TS}$ conditional on the other variables was calculated as a measure of instrument strength[111]. Briefly, MVMR by regressing a given instrumental variable on all the remaining variables as controls generates a predicted value for the instrumental variable that is not correlated with other variables in the model thus accounting for possible pleiotropic effects.

## MR phenome-wide association studies (MR-PheWAS) and PheWAS

We performed an MR-PheWAS using data from UK Biobank participants who did not undergo aortic imaging to assess the effects of aortic traits on clinical disease classifications. Using the PheWAS package (https://github.com/PheWAS/PheWAS), we mapped 1157 phecodes with more than 200 cases from the International Classification of Diseases, 9th Revision, Clinical Modification and the International Classification of Diseases, 10th Revision, Clinical Modification (ICD-10-CM)[112]. For each aortic imaging phenotype, we selected SNPs with a $p$ value $< 5 \times 10^{-8}$ and minor allele frequency >0.05 from the single-trait GWAS for AAmax, AAmin, DAmax, and DAmin, and from the multi-trait GWAS for AAdis and DAdis. Given that we considered a large number of phecodes as the outcome in the MR-PheWAS analysis with the number of cases ranging from 201 to 116,879 (median = 1353), we used a more stringent LD threshold for independent SNPs ($r^2 < 0.01$) to achieve a better-stablised model. The PheWAS model was adjusted for age, sex, genotype array, and its four principal components for population stratification. MR estimates were then obtained for each pair of aortic traits and phecode by combining the SNP-specific associations using IVW, WM, and MR-Egger. We accounted for multiple comparisons of 1157 phecodes using Bonferroni correction with $p$ value < 0.05/1157 = $4.3 \times 10^{-5}$. Standard (non-MR) PheWAS was also performed, using Python (v3.8.10) with the statsmodels (v0.12.2) package. Logistic regression was performed for each disease code against each imaging phenotype, adjusting for sex, age and BMI. The regression coefficient and two-tailed $p$ value of the imaging phenotype were reported.

## Analyses of associations of white matter hyperintensities, aortic and cervical artery dissection

LD score regression (LDSR) method was applied to test genetic correlation at the genome-wide scale for the different aortic traits with the most common MRI feature of cSVD, WMH and with CeAD. GWAS summary statistics were obtained from recently published consortia GWAS of cerebral phenotypes (from the CHARGE and CADISP consortia respectively)[44,85]. For this, common variants, mapping to the Hapmap3 reference panel were employed. As the slope from the regression of the Z score product from the two GWAS summary statistics on the LD score gives the genetic covariance, the intercept of the genetic covariance was used as an indirect measure of sample overlap[99], which corresponds to the average polygenic effects captured by genetic variants spread across the genome. A $p$ value < 0.006 (adjusting for 8 simultaneous tests) was considered significant.

LDSR could potentially miss significant correlations at the regional level due to the balancing effect[113]. A Bayesian pairwise GWAS

approach (GWAS-PW) was applied to systematically test for locally correlated regions[114]. The GWAS-PW identified trait pairs with high posterior probability of association (PPA) using a shared genetic variant (model 3, PPA3 > 0.90). To ensure that PPA3 is unbiased by sample overlap, fgwas v.0.3.6[115] was run on each pair of traits and the correlation estimated from regions with null association evidence (PPA3 < 0.20) was used as a correction factor. Additionally, to estimate the directionality of associations between trait pairs in regions with PPA3 > 0.90, a simple rank-based correlation test was applied. Independence between regions was estimated as proposed by Berisa and Pickrell[116]. Only the most strongly associated variant for the outcome per region showing high PPA3 is reported.

**Assessment of WMH in the Rhineland study**

Details on the acquisition[117], segmentation and quality assurance of WMH in the Rhineland Study have been described previously. In brief, we automatically segmented WMH using an in-house developed pipeline based on DeepMedic[118], where we utilised image information from the T1-weighted, T2-weighted, and FLAIR sequences. Estimated intracranial volume was extracted using FreeSurfer's automated segmentation (Aseg)[119]. Population characteristics have also previously been reported[117]. The total number of study participants for analysis was 3317. Genotyping strategy has been reported in detail elsewhere[120]. MVMR was applied using the same methods as described above for the main study.

**Reporting summary**

Further information on research design is available in the Nature Research Reporting Summary linked to this article.

## Data availability

The results of stage 1 GWAS and MTAG generated in this study have been deposited in the Imperial College data repository at https://doi.org/10.14469/hpc/10653 and are freely available for download. Raw data from the UKBB participants can be requested from the UKBB Access Management System (https://bbams.ndph.ox.ac.uk). Other publicly available data used for annotation and analysis are available as follows: eQTL data used in this study from aortic tissue are available at the GTEx portal (v8) (https://www.gtexportal.org/home/tissue/Artery_Aorta). HiC aorta data used for 3D chromatin interaction mapping was based on HiC aorta data (accession number GSE87112). Co-expression data used in this study are available from Gene Expression Omnibus (accession number GSE117715).

## Code availability

All the code used for the analyses is freely available. The main GWAS code (BOLT-LMM) is available for download at https://alkesgroup.broadinstitute.org/BOLT-LMM/downloads/. MTAG is available at https://github.com/JonJala/mtag.

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

## Acknowledgements

C.F. acknowledges generous support from the NHLI and Imperial College, and support from the BHF Imperial Centre of Research Excellence RE/18/4/34215 and BHF Clinical Research Fellowship FS/15/81/31817. P.M.M. acknowledges generous personal and research support from the Edmond J Safra Foundation and Lily Safra, a National Institute for Health Research (NIHR) Senior Investigator Award, the UK Dementia Research Institute and the NIHR Biomedical Research Centre at Imperial College London. This research has been conducted using the UK Biobank Resource under Application Number 18545. J.S.W. has received support from the Medical Research Council (UK), British Heart Foundation [RE/18/4/34215], NIHR Imperial College Biomedical Research Centre, Sir Jules Thorn Charitable Trust [21/JTA]. P.E. is the director of the Medical Research Council (MRC) Centre for Environment and Health and acknowledges support from the MRC (MR/L01341X/1; MR/ S019669/1). P.E. is director of the Health Protection Research Unit in Chemical and Radiation Threats and Hazards, funded by the NIHR. P.E. also acknowledges support from the NIHR Imperial Biomedical Research Centre, the Imperial College British Heart Foundation Centre for Research Excellence (RE/18/4/34215), and the UK Dementia Research Institute at Imperial College London (MC_PC_17114), and Health Data Research UK for London. S.D. is supported by a grant overseen by the French National Research Agency (ANR) as part of the'Investment for the Future' Programme ANR-18- RHUS-002 and by the ERC and the EU H2020 under grant agreements 640643, 667375, and 754517. The views expressed in this work are those of the authors and not necessarily those of the funders. The CADISP study has been supported by Inserm, Lille 2 University, Institut Pasteur de Lille and Lille University Hospital and received funding from the ERDF (FEDER funds) and Région Nord-Pas de-Calais in the frame of Contrat de Projets Etat-Region 2007-2013 Région Nord-Pas-de-Calais - Grant N°09120030, Centre National de Genotypage, Emil Aaltonen Foundation, Paavo Ilmari Ahvenainen Foundation, Helsinki University Central Hospital Research Fund, Helsinki University Medical Foundation, Päivikki and Sakari Sohlberg Foundation, Aarne Koskelo Foundation, Maire Taponen Foundation, Aarne and Aili Turunen Foundation, Lilly Foundation, Alfred Kordelin Foundation, Finnish Medical Foundation, Orion Farmos Research Foundation, Maud Kuistila Foundation, the Finnish Brain Foundation, Biomedicum Helsinki Foundation, Projet Hospitalier de Recherche Clinique Régional, Fondation de France, Génopôle de Lille, Adrinord, Basel Stroke-Funds, Käthe-Zingg-Schwichtenberg-

Fonds of the Swiss Academy of Medical Sciences, Swiss Heart Foundation. SHIP is part of the Community Medicine Research net of the University of Greifswald, Germany, which is funded by the Federal Ministry of Education and Research (grants no. 01ZZ9603, 01ZZ0103, and 01ZZ0403), the Ministry of Cultural Affairs as well as the Social Ministry of the Federal State of Mecklenburg-West Pomerania, and the network 'Greifswald Approach to Individualised Medicine (GANI_MED)' funded by the Federal Ministry of Education and Research (grant 03IS2061A). Whole-body MR imaging was supported by a joint grant from Siemens Healthineers, Erlangen, Germany and the Federal State of Mecklenburg-West Pomerania. The University of Greifswald is a member of the Caché Campus programme of InterSystems GmbH. **CADISP list of investigators:** Belgium: Departments of Neurology, Erasmus University Hospital, Brussels and Laboratory of Experimental Neurology, ULB, Brussels (Shérine Abboud, Massimo Pandolfo); Department of Neurology, Leuven University Hospial (Vincent Thijs). France: Departments of Neurology, Lille University Hospital-Inserm U1171 (Didier Leys, Marie Bodenant), Sainte-Anne University Hospital, Paris (Fabien Louillet, Emmanuel Touzé, Jean-Louis Mas), Pitié-Salpêtrière University Hospital, Paris (Yves Samson, Sara Leder, Anne Léger, Sandrine Deltour, Sophie Crozier, Isabelle Méresse), Amiens University Hospital (Sandrine Canaple, Olivier Godefroy), Dijon University Hospital (Maurice Giroud, Yannick Béjot), Besançon University Hospital (Pierre Decavel, Elizabeth Medeiros, Paola Montiel, Thierry Moulin, Fabrice Vuillier); Inserm U744, Pasteur Institute, Lille (Jean Dallongeville). Finland: Department of Neurology, Helsinki University Central Hospital, Helsinki (Antti J Metso, Tiina Metso, Turgut Tatlisumak); Germany: Departments of Neurology, Heidelberg University Hospital (Caspar Grond-Ginsbach, Christoph Lichy, Manja Kloss, Inge Werner, Marie-Luise Arnold), University Hospital of Ludwigshafen (Michael Dos Santos, Armin Grau); University Hospital of München (Martin Dichgans); Department of Rehabilitation: Schmieder-Klinik, Heidelberg (Constanze Thomas-Feles, Ralf Weber, Tobias Brandt). Italy: Departments of Neurology: Brescia University Hospital (Alessandro Pezzini, Valeria De Giuli, Filomena Caria, Loris Poli, Alessandro Padovani), Milan University Hospital (Anna Bersano, Silvia Lanfranconi), University of Milano Bicocca, San Gerardo Hospital, Monza, Italy (Simone Beretta, Carlo Ferrarese), Milan Scientific Institute San Raffaele University Hospital (Giacomo Giacolone); Department of Rehabilitation, Santa Lucia Hospital, Rome (Stefano Paolucci). Switzerland: Department of Neurology, Basel University Hospital (Philippe Lyrer, Stefan Engelter, Felix Fluri, Florian Hatz, Dominique Gisler, Leo Bonati, Henrik Gensicke, Margareth Amort). UK: Clinical Neuroscience, St George's University of London (Hugh Markus). USA: Department of Neurology, Salt Lake City, USA (Jennifer Majersik); Department of Neurology, University of Virginia, Charlottesville, USA (Bradford Worrall, Andrew Southerland); Department of Neurology, Baltimore, USA (John Cole, Steven Kittner).

## Author contributions

C.F., M.F., J.H., S.D., A.D. and P.M.M. co-designed the study. C.F., M.F., W.B., J.H., and M.S. performed quality control and core analyses. Additional data collection and analysis in cohorts for replication and secondary analyses were performed by A.T., M.B., R.B., U.V., H.V., M.D., M.-A.I., N.A.A., V.L., A.H., P.A. and S.E. S.D., E.P., J.W. and P.E. provided additional methodological guidance. C.F., M.F., J.H., A.D. and P.M.M. drafted the manuscript. All authors reviewed, contributed to serial revisions and approved the manuscript.

## Competing interests

P.M.M. acknowledges consultancy fees from Novartis, Bristol–Myers Squibb, Celgene and Biogen. He has received honoraria or speakers' honoraria from Novartis, Biogen and Roche and has received research or educational funds from Biogen, Novartis, GlaxoSmithKline and Nodthera. J.S.W. has acted as a consultant for MyoKardia and Foresite Labs, and received research support from MyoKardia & Bristol–Myers Squibb. The remaining authors declare no competing interests.

## Additional information

[1]National Heart and Lung Institute, Imperial College London, Programme in Cardiovascular Genetics and Genomics, London, UK. [2]Royal Brompton & Harefield Hospitals, Guy's and St. Thomas' NHS Foundation Trust, London SW3 6NP, UK. [3]Department of Epidemiology and Biostatistics, School of Public Health, Imperial College London, London, UK. [4]MRC London Institute of Medical Sciences (LMS), Imperial College London, London W12 0NN, UK. [5]Department of Brain Sciences, Imperial College London, London, UK. [6]Department of Computing, Imperial College London, London, UK. [7]Glenn Biggs Institute for Alzheimer's & Neurodegenerative Diseases, University of Texas Health Sciences Center, San Antonio, TX 78229, USA. [8]University of Bordeaux, Inserm, Bordeaux Population Health Research Center, team VINTAGE, UMR 1219, 33000 Bordeaux, France. [9]Institute for Community Medicine, University Medicine Greifswald, Greifswald, Germany. [10]DZHK (German Centre for Cardiovascular Research), Partner Site Greifswald, Greifswald, Germany. [11]Department of Population Medicine and Lifestyle Diseases Prevention, Medical University of Bialystok, Bialystok, Poland. [12]Population Health Sciences, German Center for Neurodegenerative Diseases (DZNE), Bonn, Germany. [13]Institute for Medical Biometry, Informatics and Epidemiology (IMBIE), Faculty of Medicine, University of Bonn, Bonn, Germany. [14]Programme in Cardiovascular & Metabolic Disorders and Centre for Computational Biology, Duke-NUS Medical School, Singapore 169857, Republic of Singapore. [15]Institute of Big Data and Artificial Intelligence, China Pharmaceutical University (CPU), 211198 Nanjing, China. [16]Computational Biology Programme, Faculty of Science, National University of Singapore, Singapore, Singapore. [17]LabEx DISTALZ-U1167, RID-AGE-Risk Factors and Molecular Determinants of Aging-Related Diseases, University of Lille, Lille, France. [18]Inserm, U1167 Lille, France. [19]Centre Hospitalier Universitaire Lille, Lille, France. [20]Institut Pasteur de Lille, Lille, France. [21]Department of Neurology and Stroke Center, University Hospital and University of Basel, Petersgraben 4, CH – 4031 Basel, Switzerland. [22]Department of Clinical Neurology and Neurorehabilitation, University Department of Geriatric Medicine FELIX PLATTER, University of Basel, Basel, Switzerland. [23]Department of Radiology and Neuroradiology, University Medicine Greifswald, Greifswald, Germany. [24]Interfaculty Institute for Genetics and Functional Genomics, University Medicine Greifswald, Greifswald, Germany. [25]Department of Internal Medicine B, University Medicine Greifswald, Greifswald, Germany. [26]Department of Neurology, Faculty of Medicine, University of Bonn, Bonn, Germany. [27]Department of Neurology, Institute for Neurodegenerative Diseases, Bordeaux University Hospital – CHU Bordeaux, 33000 Bordeaux, France. [28]UK Dementia Research Institute at Imperial College London, London, UK. [29]Health Data Research (HDR) UK London at Imperial College London, London, UK. [30]Britsh Heart Foundation Centre of Research Excellence at Imperial College London, London, UK. [31]National Institute for Health Research Imperial Biomedical Research Centre, Imperial College London, London, UK. [32]MRC Centre for Environment and Health, School of Public Health, Imperial College London, London, UK. [33]These authors contributed equally: Catherine M. Francis, Matthias E. Futschik, Jian Huang. [34]These authors jointly supervised this work: Abbas Dehghan, Paul M. Matthews. ✉e-mail: a.dehghan@imperial.ac.uk; p.matthews@imperial.ac.uk

