## [Peer Review File · Nature Communications]

GENOME-WIDE ASSOCIATIONS OF AORTIC DISTENSIBILITY SUGGEST CAUSALITY FOR AORTIC ANEURYSMS AND BRAIN WHITE MATTER HYPERINTENSITIESREVIEWER COMMENTS

Reviewer #1 (Remarks to the Author):

This manuscript leverages MRI imaging in the UK Biobank (n~32K white individuals) to generate 6 quantitative thoracic aortic traits, including aortic distensibility. The authors performed GWAS and identified ~100 loci, with many harboring genes in the TGF- β , IGF, VEGF and PDGF pathways. Mendelian randomization provided evidence for a causal role for aortic distensibility in development of aortic aneurysms and cerebral white matter hyperintensities (a marker of small vessel disease).

Specific comments:

1. Early in the results section the authors should comment on the relationship between distensibility (and other aortic measures) and demographic/clinical factors including age, sex, blood pressure, diabetes, etc. A supplementary table showing the basic demographic/clinical features of the cohort studies and the relationship to distensibility would be helpful.
2. All non-white individuals were excluded from this analysis. Were the aortic measures quantitated in these individuals? It would be of interest to report on them even though they were not included in the GWAS analyses.
3. The GWAS lacks replication. Are there any different cohorts with aortic imaging and genotyping that could be utilized for replication?
4. The UK Biobank also has publicly available whole exome data. What is the overlap of these individuals with the ~32K used for aortic measures? The authors should consider examining genes nominated by their GWAS for coding variants detected by exome sequencing that are associated, singly or via gene burden analysis, with the same aortic trait.
5. The authors should clarify the directionality of the effects of the TGF- β , IGF, PDGF and VEGF signaling pathways on aortic distensibility. For example, do the aggregated data support that increased or decreased TGF- β signalling is associated with lower aortic distensibility? This should be clarified for all the major pathways identified by these analyses, as well as key genes such as elastin and other ECM proteins. Here again, an analysis of a burden of pLOFs in selected genes from the exome sequencing data could help to support inferences around directionality.

Reviewer #2 (Remarks to the Author):

1. Table 1 should report the other allele as well as the effect allele. The effect allele frequency should also be reported rather than the MAF.
2. Did the applicants perform power calculations for each of six traits included in the analyses? Both overall and sex-specifically?
3. Table 2 should report the other allele as well as the effect allele. The effect allele frequency should also be reported.
4. Line 380 -381 – why didn't the applicants adjust for array design and principal components in these analyses as well as for age and sex?
5. It is well known that UK biobank is not representative of the UK population. Could selection bias have influenced any of these results and did the authors take any steps to mitigate the potential for selection bias to affect the findings?
6. Lines 540-549: The findings of associations between aortic traits and small vessel disease are very interesting. Did the authors consider replication of these findings in an external cohort?
7. Lines 601-602: What was the accuracy of the CNN? Are there kappa statistics and other measures of reliability and accuracy reported?
8. Line 615: Details of what sort of QC was performed should be given and the results should be reported in the supplementary materials.
9. Lines 630-634: Why didn't the authors adjust for array design and principal components in these

analyses?

10. Lines 642-646: the MTAG methodology is not very clearly described – I assume it is some form of joint modelling approach. Can the authors provide more detail on how this method increases power?

11. Lines 732-735: The authors used the contamination mixture method to test the validity of instrumental variables. Did they consider other approaches such as MR-GRAPPLE that is used for two-sample MR with heterogeneous instruments as well?

12. Lines 791-792: The authors state that “The PheWAS model was adjusted for age, sex, genotype array, and its four principal components for population stratification.” Why are genotype array adjusted and PCs adjusted for in these analyses and not in others? Why were only 4 PCs and not at least 10 PCs out of the 40 PCs available in UKB?

We wish to thank the reviewers for their careful consideration of our data and their insightful comments. We address specific points and questions raised by the reviewers below with all revisions in the manuscript text highlighted in red font.

We are also aware of an additional paper which has been published since this research was submitted (Benjamins et al¹), which reports some GWAS data for ascending aortic distensibility, and aortic areas. They report only four loci reaching genome-wide significance for ascending aortic distensibility. Using more accurate phenotyping (see: Dice metrics in Methods) and joint modelling (MTAG), we identify an additional 22 loci associated with ascending aortic distensibility and 13 additional loci for descending aortic distensibility, adding significantly to the understanding of these clinically relevant phenotypes. We have incorporated the new publication into our comparative table to highlight which of our associations are novel.

We have demonstrated reproducibility of our data (both stage 1 GWAS and MTAG “hits”) by replicating effect directions for aortic areas in a smaller population cohort. Our exploratory analysis of the possible causal relationship between aortic traits and white matter hyperintensities also adds to the novelty of our findings.

REVIEWER COMMENTS

Reviewer #1 (Remarks to the Author):

This manuscript leverages MRI imaging in the UK Biobank (n~32K white individuals) to generate 6 quantitative thoracic aortic traits, including aortic distensibility. The authors performed GWAS and identified ~100 loci, with many harboring genes in the TGF- β , IGF, VEGF and PDGF pathways. Mendelian randomization provided evidence for a causal role for aortic distensibility in development of aortic aneurysms and cerebral white matter hyperintensities (a marker of small vessel disease).

Specific comments:

1. Early in the results section the authors should comment on the relationship between distensibility (and other aortic measures) and demographic/clinical factors including age, sex, blood pressure, diabetes, etc. A supplementary table showing the basic demographic/clinical features of the cohort studies and the relationship to distensibility would be helpful.

Thank you for this suggestion. This is indeed interesting and important information. We have added a Supplementary Table 1c for the relationships of demographic and clinical factors such as age, BMI, diabetes and hypertension to aortic traits, and we have added comments on these relationships to the Results section.

2. All non-white individuals were excluded from this analysis. Were the aortic measures quantitated in these individuals? It would be of interest to report on them even though they were not included in the GWAS analyses.

We did indeed measure the aortic traits in the non-white individuals, and now have reported this data for interest in the supplementary information. The data is now included in Supplementary figure S2C and Supplementary table S1b

3. The GWAS lacks replication. Are there any different cohorts with aortic imaging and genotyping that could be utilized for replication?

We have used a smaller population cohort (SHIP) which also included both genotyping and MRI data (N=2787). Whilst underpowered for discovery, the additional data has allowed us to demonstrate replication in an independent cohort, with consistent effect direction in >89% of the overlapping lead SNPs for aortic dimensions, both in our stage 1 GWAS and our MTAG analysis. This has confirmed the robustness and validity of the MTAG analysis findings, as well as the stage 1 GWAS results. This replication data has been added to the manuscript (with detailed methods for the SHIP cohort in a new Supplementary Information file), and look-ups are included in Supplementary Tables 7a-d.

We were not able to find a sufficiently large cohort to allow direct replication of our associations with distensibility (distensibility data was not acquired as part of the SHIP study), but as distensibility is a phenotype derived from measures of aortic dimensions, we believe that the robustness of our reported associations is supported by the replication data for aortic dimensions described above. Note that we approached the investigators of the only other large population study (MESA) that we are aware of for which MRI-derived distensibility data was obtained, but they were unable to share their data with us at this time.

4. The UK Biobank also has publicly available whole exome data. What is the overlap of these individuals with the ~32K used for aortic measures? The authors should consider examining genes nominated by their GWAS for coding variants detected by exome sequencing that are associated, singly or via gene burden analysis, with the same aortic trait.

The overlap of subjects at present is 18,571. Exploration of rare variant associations in the available whole exome data is beyond the scope of this paper. We agree that it would be of value and will assemble a group for this purpose as part of future research plans, but are awaiting the release from UK Biobank of the full 450,000+ exomes (which will allow full overlap with our aortic traits cohort) before undertaking this, to ensure a well-powered study.

5. The authors should clarify the directionality of the effects of the TGF- β , IGF, PDGF and VEGF signaling pathways on aortic distensibility. For example, do the aggregated data support that increased or decreased TGF- β signalling is associated with lower aortic distensibility? This should be clarified for all the major pathways identified by these analyses, as well as key genes such as elastin and other ECM proteins. Here again, an analysis of a burden of pLOFs in selected genes from the exome sequencing data could help to support inferences around directionality.

The reviewer asks an important question, but one which is extremely difficult to address. Not only is inferring the effect of a detected variant on the activity of a mapped pathway gene by itself a formidable task, the complexity of signalling pathways makes any general assessment of their signalling activity highly challenging, particularly where not only organ-specific but also lineage-specific effects on signalling pathways are seen to play an important role in phenotype. Nevertheless, to gain initial insights, we utilised information from GTEX for the detected eQTLs to putatively link gene expression to the phenotypic variables (aortic area and distensibility) in our study. Aligning the directionality of the effects (beta) in our study with the directionality of expression changes in GTEX for aortic tissue enabled us to predict associations between the expression of candidate genes and the phenotypic measures. While we could align over 8000 eQTLs mapped to 164 genes, the coverage of signalling pathway and ECM remains limited. Furthermore, this mapping was only possible for candidate genes with mapped significant eQTLs, which excluded some interesting candidates such as elastin. Despite these limitations, the exploratory analysis suggested relationships of interest between for example, distensibility and genes known to influence TGF- β signalling (such as positively signed correlations between distensibility and FGF9 or THSD4

expression). Although these exploratory data warrant cautious interpretation present, we present them in a supplementary figure S13 and supplementary table S15c.

As noted above, we felt that investigation of the WES – and potential LOF associations- was beyond the scope of this paper. We plan to examine this once the full 450+ exomes (allowing full overlap with this dataset for a well-powered analysis) have been released by UK Biobank.

Reviewer #2 (Remarks to the Author):

1. Table 1 should report the other allele as well as the effect allele. The effect allele frequency should also be reported rather than the MAF.

This has now been amended

2. Did the applicants perform power calculations for each of six traits included in the analyses? Both overall and sex-specifically?

We performed power calculations using assumptions derived from previously published studies which suggest that standardised effect sizes for aortic traits average around 0.05. The largest effect sizes are around 0.2. Power heatmaps are provided in a new Supplementary Figure S18 for a range of MAF and standardised betas, given our cohort sizes (whole cohort and sex-specific). Calculation using *gwas-power* in R (which uses the formulae in Appendix 1 of Visscher et al²) and checked with *Quanto* (<https://pphs.usc.edu/download-quanto/>) suggest that the power with our full cohort size of 32590 for a beta of 0.05 at a MAF of 0.3 is 0.66 to detect genome-wide significant associations at $p < 5 \times 10^{-8}$. Increasing the sample size by 10,000 individuals would increase our power to detect a beta of 0.05 at a MAF of 0.3, to 0.89. These figures motivated us to increase power using MTAG as described in the paper.

3. Table 2 should report the other allele as well as the effect allele. The effect allele frequency should also be reported.

This has now been included.

4. Line 380 -381 – why didn't the applicants adjust for array design and principal components in these analyses as well as for age and sex?

We included adjustment for array design and 4PCs in the MR-PheWAS analysis – this was an omission in the text, now corrected.

5. It is well known that UK biobank is not representative of the UK population. Could selection bias have influenced any of these results and did the authors take any steps to mitigate the potential for selection bias to affect the findings?

We agree that the UK Biobank is not representative of the UK population and has a well-recognised “healthy volunteer” selection bias. However, genotype -phenotype association should be generalisable, as they do not rely on the assumption of population representativeness. We have added a note to the limitations section of the discussion to this effect:

“The GWAS was restricted to the analysis of Caucasian individuals and additionally it is well-recognised that UK Biobank is not representative of the UK population as a whole³. While genotype-phenotype associations can be biased by population stratification, our analysis was adjusted for ethnicity and relatedness. Nonetheless, the “healthy volunteer” selection bias of the cohort could be a potential confound if it significantly influenced the aortic traits of interest⁴. To address this potential confound we now have demonstrated replication in an independent European population cohort (SHIP) for our primary GWAS findings and have shown that the findings in UK Biobank can be used to predict related cardiovascular traits (WMH) in another independent cohort (CHARGE).”

6. Lines 540-549: The findings of associations between aortic traits and small vessel disease are very interesting. Did the authors consider replication of these findings in an external cohort?

We attempted replication in the only independent cohort for which we had access to both imaging-derived WMH and genotyping data (the Rhineland study). The analysis was underpowered with $N=3317$ (as compared with $N>50,000$ for the primary analysis using the CHARGE cohort); the conditional f-stat was very low ($F<10$) for the MVMR (see new Supplementary Table S25). We were therefore not able fully to replicate our primary findings. We noted that effect directions were consistent for the distensibility traits, but not for the area traits. We therefore have characterised the associations of aortic traits with WMH that we report as exploratory analyses. This data that we used in our attempt to replicate has been added to the manuscript.

7. Lines 601-602: What was the accuracy of the CNN? Are there kappa statistics and other measures of reliability and accuracy reported?

In terms of segmentation accuracy, the neural network achieves a Dice metric of 0.960 for the ascending aorta and 0.953 for the descending aorta. The Dice metric is a commonly used metric for evaluating image segmentation accuracy. A Dice metric of over 0.95 is typically regarded as of high accuracy. Furthermore, the aortic segmentations underwent an automated quality control step and a manual checking step, as explained in the response to Line 615. This data has now been added to the manuscript.

8. Line 615: Details of what sort of QC was performed should be given and the results should be reported in the supplementary materials.

In detail, the automated segmentation was quality controlled using the following criteria: 1) the aorta appears in all the time frames of the image sequence; 2) there is no abrupt change of aortic areas between adjacent time frames; 3) the aortic segmentation should be a single connected component. Any segmentation that did not fulfil these criteria were excluded in the quality control. Subsequently, an image analyst visually assessed the segmentation screenshot. The aortic images were available for 37,891 subjects. After image analysis and quality control, the imaging phenotypes were available for 36,995 subjects.

9. Lines 630-634: Why didn't the authors adjust for array design and principal components in these analyses?

Here, we did not adjust for principal components as we used BOLT-LMM linear mixed modelling, which allows correction for population stratification and relatedness.

The array design was not included as exploratory analysis suggested that there was no inflation of lambda nor effect on the lead SNPs reported with inclusion of this as a covariate. We re-ran a full GWAS of aortic area including array design as a covariate, and lambda was unaffected, as were the lead SNPs – so this would not affect the reported loci (this data can be made available upon request).

10. Lines 642-646: the MTAG methodology is not very clearly described – I assume it is some form of joint modelling approach. Can the authors provide more detail on how this method increases power?

MTAG has been described in detail in Turley et al⁵ and used across many published GWAS studies including from our group to boost power⁶⁻⁸. The basic idea is that when there is correlation between GWAS estimates from different traits, it is possible to improve the accuracy and power of the effect estimates for each individual trait by including information contained in the GWAS estimates for the other traits. It is a generalisation of inverse-variance-weighted meta-analysis, assuming a constant variance:covariance matrix of effect sizes across traits. This data and the MTAG reference paper has been added to the manuscript

11. Lines 732-735: The authors used the contamination mixture method to test the validity of instrumental variables. Did they consider other approaches such as MR-GRAPPLE that is used for two-sample MR with heterogeneous instruments as well?

MR-GRAPPLE was not available at the time we planned this study. The technique uses a flexible p-value threshold for instrument selection to avoid weak instrument bias, particularly for multiple risk factors, based on the profile likelihood framework⁹. It claims to address pervasive pleiotropy and correlated risk factors in an integrated framework.

We have however used MR contamination mixture method, also a likelihood-based method, for obtaining valid causal inferences with some invalid instrumental variables¹⁰. The two methods serve similar purposes. In this study, we applied contamination mixture method to MR with a single target risk factor, as our group has previously applied this method. We are unaware of any study that has independently compared these two methods, and therefore it is hard to know which one is superior. Both methods appear to be valid. If the editor finds it necessary, we are happy to run MR-GRAPPLE.

12. Lines 791-792: The authors state that “The PheWAS model was adjusted for age, sex, genotype array, and its four principal components for population stratification.” Why are genotype array adjusted and PCs adjusted for in these analyses and not in others? Why were only 4 PCs and not at least 10 PCs out of the 40 PCs available in UKB?

We adjusted for PCs and genotype array here as the technique used for association in our PheWAS does not account for population stratification and relatedness (as BOLT-LMM does in the main GWAS analyses). We have restricted our analysis to the Caucasian participants, thus we did not adjust for more PCs. Our group has previously performed PheWAS adjusting for 4 PCs and found it sufficient for this avoid population stratification whilst not introducing more error¹¹.

We believe that these revisions have made this a stronger paper and, in particular that the replication supports the biological and clinical relevance of our findings. We hope that we have

addressed these comments in a satisfactory manner and are grateful for your renewed consideration of the paper with the additional data and revisions.

1. Benjamins JW, Yeung MW, van de Vegte YJ, Said MA, van der Linden T, Ties D, et al. Genomic insights in ascending aortic size and distensibility. *EBioMedicine*. 2022;75:103783.
2. Visscher PM, Wray NR, Zhang Q, Sklar P, McCarthy MI, Brown MA, et al. 10 Years of GWAS Discovery: Biology, Function, and Translation. *American journal of human genetics*. 2017;101(1):5-22.
3. Fry A, Littlejohns TJ, Sudlow C, Doherty N, Adamska L, Sprosen T, et al. Comparison of Sociodemographic and Health-Related Characteristics of UK Biobank Participants With Those of the General Population. *American journal of epidemiology*. 2017;186(9):1026-34.
4. Munafo MR, Tilling K, Taylor AE, Evans DM, Davey Smith G. Collider scope: when selection bias can substantially influence observed associations. *Int J Epidemiol*. 2018;47(1):226-35.
5. Turley P, Walters RK, Maghzian O, Okbay A, Lee JJ, Fontana MA, et al. Multi-trait analysis of genome-wide association summary statistics using MTAG. *Nature genetics*. 2018;50(2):229-37.
6. Tadros R, Francis C, Xu X, Vermeer AMC, Harper AR, Huurman R, et al. Shared genetic pathways contribute to risk of hypertrophic and dilated cardiomyopathies with opposite directions of effect. *Nature genetics*. 2021;53(2):128-34.
7. Kalra G, Milon B, Casella AM, Herb BR, Humphries E, Song Y, et al. Biological insights from multi-omic analysis of 31 genomic risk loci for adult hearing difficulty. *PLoS genetics*. 2020;16(9):e1009025.
8. Lee JJ, Wedow R, Okbay A, Kong E, Maghzian O, Zacher M, et al. Gene discovery and polygenic prediction from a genome-wide association study of educational attainment in 1.1 million individuals. *Nature genetics*. 2018;50(8):1112-21.
9. Wang J, Zhao Q, Bowden J, Hemani G, Davey Smith G, Small DS, et al. Causal inference for heritable phenotypic risk factors using heterogeneous genetic instruments. *PLoS genetics*. 2021;17(6):e1009575.
10. Burgess S, Foley CN, Allara E, Staley JR, Howson JMM. A robust and efficient method for Mendelian randomization with hundreds of genetic variants. *Nature communications*. 2020;11(1):376.
11. Gill D, Benyamin B, Moore LSP, Monori G, Zhou A, Koskeridis F, et al. Associations of genetically determined iron status across the phenome: A mendelian randomization study. *PLoS Med*. 2019;16(6):e1002833.

REVIEWERS' COMMENTS

Reviewer #1 (Remarks to the Author):

None

Reviewer #2 (Remarks to the Author):

I am happy with the responses to my comments.

REVIEWERS' COMMENTS

Reviewer #1 (Remarks to the Author):

None

Reviewer #2 (Remarks to the Author):

I am happy with the responses to my comments.

Response: we are pleased to have addressed the reviewers' previous comments and thank them for their careful consideration and oversight of our work.

We include previous comments and responses below for completeness.